# LncRNA *RUS* shapes the gene expression program towards neurogenesis

Marius F Schneider[1,2] , Veronika Müller[2], Stephan A Müller[3,4], Stefan F Lichtenthaler[3,4,5] , Peter B Becker[1] , Johanna C Scheuermann[2]

The evolution of brain complexity correlates with an increased expression of long, noncoding (lnc) RNAs in neural tissues. Although prominent examples illustrate the potential of lncRNAs to scaffold and target epigenetic regulators to chromatin loci, only few cases have been described to function during brain development. We present a first functional characterization of the lncRNA *LINC01322*, which we term *RUS* for "RNA upstream of Slitrk3." The *RUS* gene is well conserved in mammals by sequence and synteny next to the neurodevelopmental gene Slitrk3. *RUS* is exclusively expressed in neural cells and its expression increases during neuronal differentiation of mouse embryonic cortical neural stem cells. Depletion of *RUS* locks neuronal precursors in an intermediate state towards neuronal differentiation resulting in arrested cell cycle and increased apoptosis. *RUS* associates with chromatin in the vicinity of genes involved in neurogenesis, most of which change their expression upon *RUS* depletion. The identification of a range of epigenetic regulators as specific *RUS* interactors suggests that the lncRNA may mediate gene activation and repression in a highly context-dependent manner.

## Introduction

Most parts of a higher eukaryotic genome are transcribed at times and in certain cells, but only a minority of the resulting RNAs are protein-coding. Whereas many of these noncoding transcripts are immediately degraded, others are processed into small RNAs that form an intricate network regulating gene expression in a co- and post-transcriptional manner. In addition, mammalian genomes encode thousands of stable RNAs longer than 200 nucleotides, often capped and polyadenylated, but without any obvious coding potential (long, noncoding [lnc] RNAs) (Engreitz et al, 2016; Quinn & Chang, 2016; Rutenberg-Schoenberg et al, 2016; Kopp & Mendell, 2018). The functions of most lncRNAs discovered in large-scale

sequencing projects remain to be explored. "Guilt-by-association" strategies correlate their presence and expression levels with certain cellular states, including disease conditions. Increasingly, interference strategies reveal critical roles for lncRNAs in cellular fates and states (Lin et al, 2014; Rinn & Chang, 2020; Statello et al, 2021).

Apparently, lncRNAs arise by pervasive transcription of the genome and evolve fast. Conceivably, their structural flexibility makes them an ideal substrate for "constructive neural evolution" and predisposes them for a function in chromatin regulation (Palazzo & Koonin, 2020; Rinn & Chang, 2020). Indeed, more than 60% of annotated lncRNAs in human cells are chromatin-enriched (Rinn & Chang, 2012). In the chromatin context, lncRNAs often combine two functions: scaffolding and targeting. The intrinsic ability of lncRNAs to mediate positional targeting in the genome qualifies them to impose allele-specific epigenetic regulation, such as genome imprinting, X chromosome inactivation or rDNA regulation (Yao et al, 2019; Rinn & Chang, 2020; Statello et al, 2021). Their actions may be locally restricted close to their site of transcription in *cis*, or in *trans* via sequence-specific hybridization with DNA or RNA. Thus, they may guide powerful "epigenetic" regulators (enzymes that modify histones or DNA) to specific loci in chromatin, or participate in nuclear condensates (Engreitz et al, 2016; Rutenberg-Schoenberg et al, 2016; Kopp & Mendell, 2018; Statello et al, 2021). Prominent examples of lncRNAs recruiting regulators that define epigenetic chromatin states, include *XIST*, *HOTAIR*, and *ANRIL* that bind polycomb complexes (PRC) to silence chromosomal regions, whereas others such as HOTTIP or certain enhancer RNAs are known to recruit activating histone acetyltransferase or methylase complexes (Werner & Ruthenburg, 2015; Quinn & Chang, 2016).

The fraction of lncRNAs that are expressed in a tissue-specific manner exceeds that of cell type-specific protein-coding genes (Djebali et al, 2012). A particular rich compendium of lncRNAs is expressed in the mammalian brain (estimated 40% of known lncRNAs) (Mercer et al, 2010; Briggs et al, 2015; Hezroni et al, 2019), and a strong correlation between the number of expressed lncRNAs and mammalian brain size was reported (Clark & Blackshaw, 2017).

[1]Division of Molecular Biology, Biomedical Center Munich, Ludwig-Maximilians-University, Munich, Germany   [2]Division of Metabolic Biochemistry, Faculty of Medicine, Biomedical Center Munich (BMC), Ludwig-Maximilians-Universität München, Munich, Germany   [3]Neuroproteomics, School of Medicine, Klinikum rechts der Isar, Technical University of Munich, Munich, Germany   [4]German Center for Neurodegenerative Diseases (DZNE) Munich and Neuroproteomics Unit, Technical University, Munich, Germany   [5]Munich Cluster for Systems Neurology (SyNergy), Munich, Germany

Correspondence: pbecker@bmc.med.lmu.de

Brain-specific lncRNAs tend to be more evolutionary conserved between orthologues than lncRNAs expressed in other tissues and their genes often reside next to protein-coding genes involved in neuronal development or brain function processes (Ponjavic et al, 2009). Indeed, lncRNAs are drivers of key neurodevelopmental processes such as neuroectodermal lineage commitment, proliferation of neural precursor cells, specification of the precursor cells, and the differentiation of precursor cells into neurons (neurogenesis) or other neural cell types (gliogenesis) (Briggs et al, 2015; Zimmer-Bensch, 2019).

Diverse mechanisms have been documented. For example, lncRNA *TUNA* (*megamind*) is involved in neural differentiation of mouse embryonic stem cells (Lin et al, 2014). The finding that depletion of *TUNA* also compromised ESC proliferation and maintenance of pluripotency illustrates the power of lncRNA to control gene networks in diverse ways, depending on the nature of protein effectors and the timing and context of their lncRNA interactions (Lin et al, 2014). The lncRNA *RMST* promotes neuronal differentiation by recruiting the transcription factor Sox2 to promoters of neurogenic genes (Ng et al, 2013). The lncRNA *Pinky* is expressed in the neural lineage, where it helps to maintain the proliferation of a transit-amplifying cell population, thereby restraining neurogenesis. This regulation takes place at the level of transcript splicing, illustrating the versatility of nuclear lncRNAs (Ramos et al, 2015). Other mechanisms involve the control of miRNA availability and function, as has been shown for the primate-specific lncND during neurodevelopment (Rani et al, 2016).

Only a small fraction of lncRNAs involved in neurodevelopment and brain function has been studied in detail. We here describe a novel lncRNA involved in neurogenesis, which we term *RUS* (for "RNA upstream of Slitrk3"). The *RUS* gene resides at a syntenic position in mouse and human genomes upstream of the Slitrk3 gene, which encodes a transmembrane protein involved in suppressing neurite outgrowth. *RUS* is expressed in neural tissues only and its expression increases during the differentiation of neural stem cells (NSCs) into neurons. *RUS* is a nuclear lncRNA that interacts with chromatin in the vicinity of genes involved in neurogenesis. Depletion of RUS results in massive alterations in the gene expression program of neuronal progenitor cells, trapping them in an intermediate state during differentiation and eventually leading to proliferation arrest. Proteomic identification of *RUS*-interacting proteins suggests multiple mechanisms of *RUS*-mediated epigenetic gene regulation.

## Results

### Identification of the neuronal-specific lncRNA RUS

To identify novel, functionally relevant lncRNAs in the context of neurogenesis, we took advantage of prior work of Ziller et al, who profiled transcription during differentiation of human embryonic stem cells along the neural lineage (Ziller et al, 2015). Their data include transcriptome profiles of hESC-derived neural progenitors: neuroepithelial cells (NE), early, mid and late radial glia cells (ERG, MRG, and LRG, respectively) and their in vitro differentiated counterparts (Ziller et al, 2015). We evaluated 553 candidate lncRNA

transcripts according to the following criteria. They should (1) only be expressed in neural tissues, (2) be dynamically regulated during the differentiation of neural precursor cells, and (3) be conserved between mouse and humans (Fig 1A). Of these, 10 transcripts decrease and 29 increase during the differentiation of the four cell types (Fig 1B). Among them, we identified *LINC01322* as an interesting candidate, as it was absent in NE, ERG, and MRG but expressed in all differentiated cell types. Intriguingly, *LINC01322* was also expressed in undifferentiated LRG.

LncRNA genes relevant to neurogenesis are often located next to neurodevelopmental protein-coding genes (Ponjavic et al, 2009). In line with this observation, the gene for *LINC01322* localizes upstream of the gene encoding the transmembrane protein Slitrk3, which regulates neurite outgrowth (Aruga et al, 2003) (Fig 1C). In the following, we refer to *LINC01322* as *RUS* (*RNA upstream to Slitrk3*). The location of the *RUS* gene is well conserved by synteny in mice and humans between the *Slitrk3* and *Bche-201* genes (Fig 1C).

The murine *RUS* transcript, *Gm20754*, has two annotated isoforms. Two and five exons are annotated for isoforms 1 and 2, respectively. Both isoforms share the 232 bp exon 1, which is 75% similar to the orthologous counterpart in humans (Fig 1C). The sequence of *mRUS* exon 2 (114 bp) is conserved to 92%, but not part of the predominant human transcript. In silico ORF predictions revealed that the largest ORF encodes a theoretical polypeptide of 80 amino acids (aa). Although the corresponding peptides are not listed in the comprehensive peptide repository (http://www.peptideatlas.org), we cannot exclude a functional role for a hypothetical polypeptide encoded by this small ORF. Likewise, we cannot exclude that *RUS* is processed to miRNAs (https://www.mirbase.org/) contributing to its functionality.

Quantitative RT-PCR (RT-qPCR) analysis of the two isoforms in different mouse adult and embryonic tissues revealed that *RUS* annotated isoform 1 is the dominant form (Fig 1D). *RUS* expression is restricted to neural tissues, with highest expression in the adult hippocampus. We further explored the spatio-temporal expression of *RUS* isoform 1 in the developing mouse brain. RT-qPCR analyses of *RUS-1* transcripts in cortex and hippocampus of different developmental stages (embryonic days E14 and E18, postnatal days P3 and P8, as well as adult animals) showed that *RUS-1* expression increased during cortical development and peaked on P3 when nestin, a marker for neural precursor cells dropped. A reciprocal expression pattern was observed in the hippocampus. Continuing with isoform 1, we performed 3′-RACE experiments to obtain the annotated 3′ end (Fig S1A). However, amplification of *RUS* with primers targeting the annotated 5′ and 3′ ends yielded two PCR bands of 1.3 and 0.9 kbp. Sequencing the more abundant 0.9-kbp PCR band revealed that it lacked exon 4 (Fig S1B).

### RUS depletion leads to reduced neuronal differentiation, proliferation arrest, and increased apoptosis

To monitor the expression of *RUS* during murine neurogenesis, we differentiated embryonic cortical neural stem cells (NSCs) into immature neurons in vitro (Kilpatrick & Bartlett, 1993; Azari et al, 2011; Mukhtar et al, 2020). Differentiating NSCs were maintained proliferative by mitogen (bFGF) for the first 4 d. On day 5, bFGF was withdrawn to induce neurogenesis (Fig S2A). During a time course of

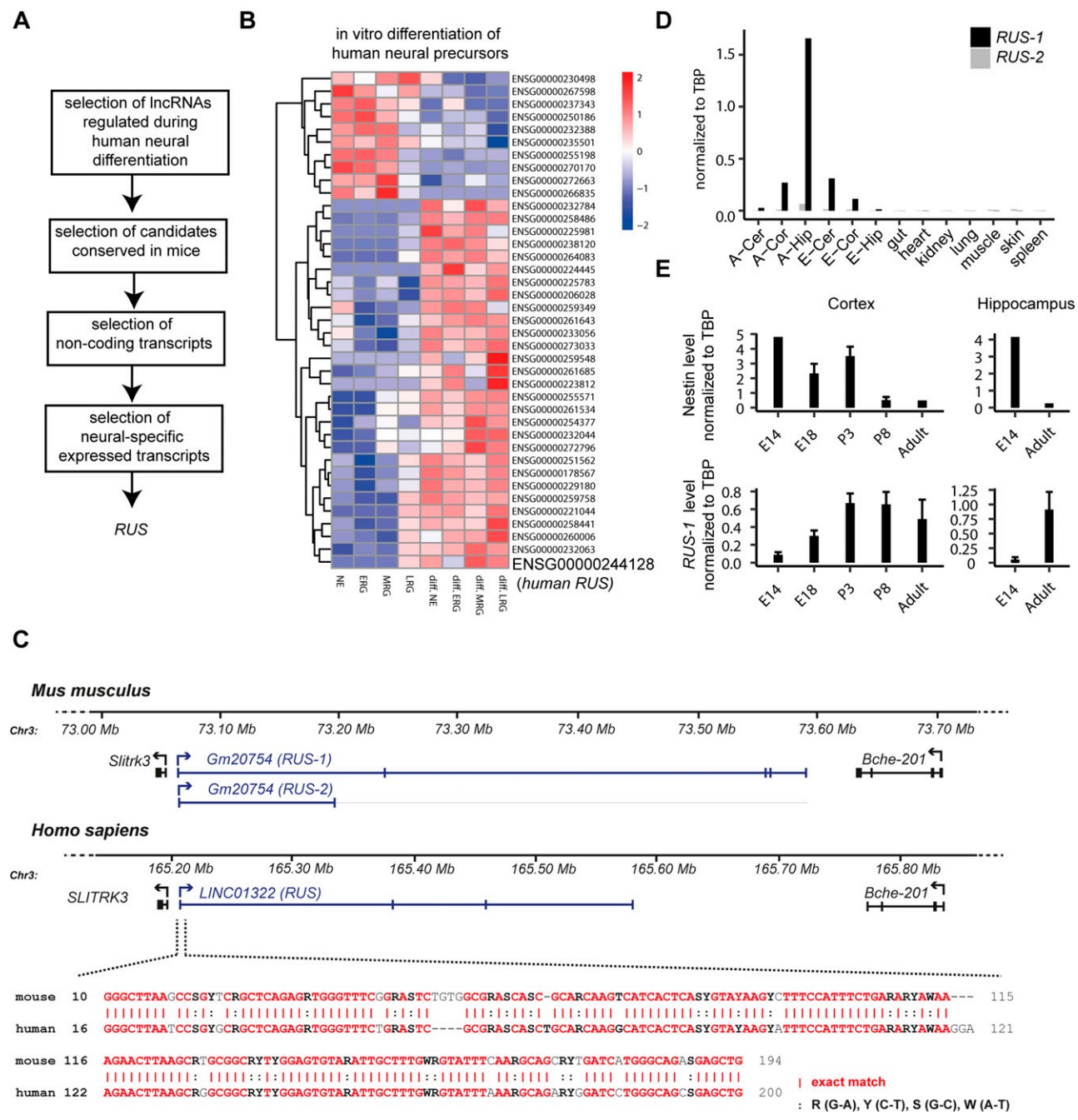

**Figure 1. RUS is a novel, conserved lncRNA involved in neurogenesis.**
**(A)** Workflow illustrating the criteria to identify candidate lncRNAs expressed in human ESC-derived NE, ERG, MRG, and LRG before and after differentiation in the data of (Ziller et al, 2015). This led to the selection of the conserved lncRNA *RUS* as subject of this study performed in mouse cells. **(B)** Heat map of significantly changed lncRNAs expressed in human ESC-derived NE, ERG, MRG, and LRG before and after differentiation (two-sided *t* test). Data of Ziller et al (2015) were analyzed. **(C)** Conservation of the *RUS* gene between mouse and human genomes by synteny (top) and by sequence of exon 1 (bottom). Note that the RUS gene resides just upstream of the *Slitrk3* gene in either case. For mice, two *RUS* isoforms are indicated. **(C, D)** Expression of murine *RUS-1* and *RUS-2* isoforms (see panel C) in different embryonic (E-) and adult (A-) tissues: cortex (Cor), cerebellum (Cer), hippocampus (Hip), gut, heart, kidney, liver, lung, muscle, skin, spleen, analyzed by RT-qPCR. **(E)** Expression of *Nestin* and *RUS* isoform 1 in murine cortex and hippocampus at different developmental stages: embryonic day (E) 14 (n = 1 for *Nestin*, n = 2 for *RUS*), E18 (n = 5), postnatal day (P) 3 (n = 2), P8 (n = 4), and in the adult mouse (n = 1 for *Nestin*, n = 2 for *RUS*). Error bars show the standard error of the mean. The values were normalized to expression constitutive *TBP* mRNA (arbitrary units, in D and E).

9 d, the expected changes in molecular marker expression were detected via immunostaining and RT-qPCR analyses. The high expression of the NSC marker *Nestin* decreased, with a concomitant increase in RGC markers *Gfap*, *Glast*, and *GluL* (Figs 2A and S2A and B), as observed elsewhere (Imura et al, 2003; Mamber et al, 2012). Upon bFGF withdrawal, the culture acquired neuronal features with

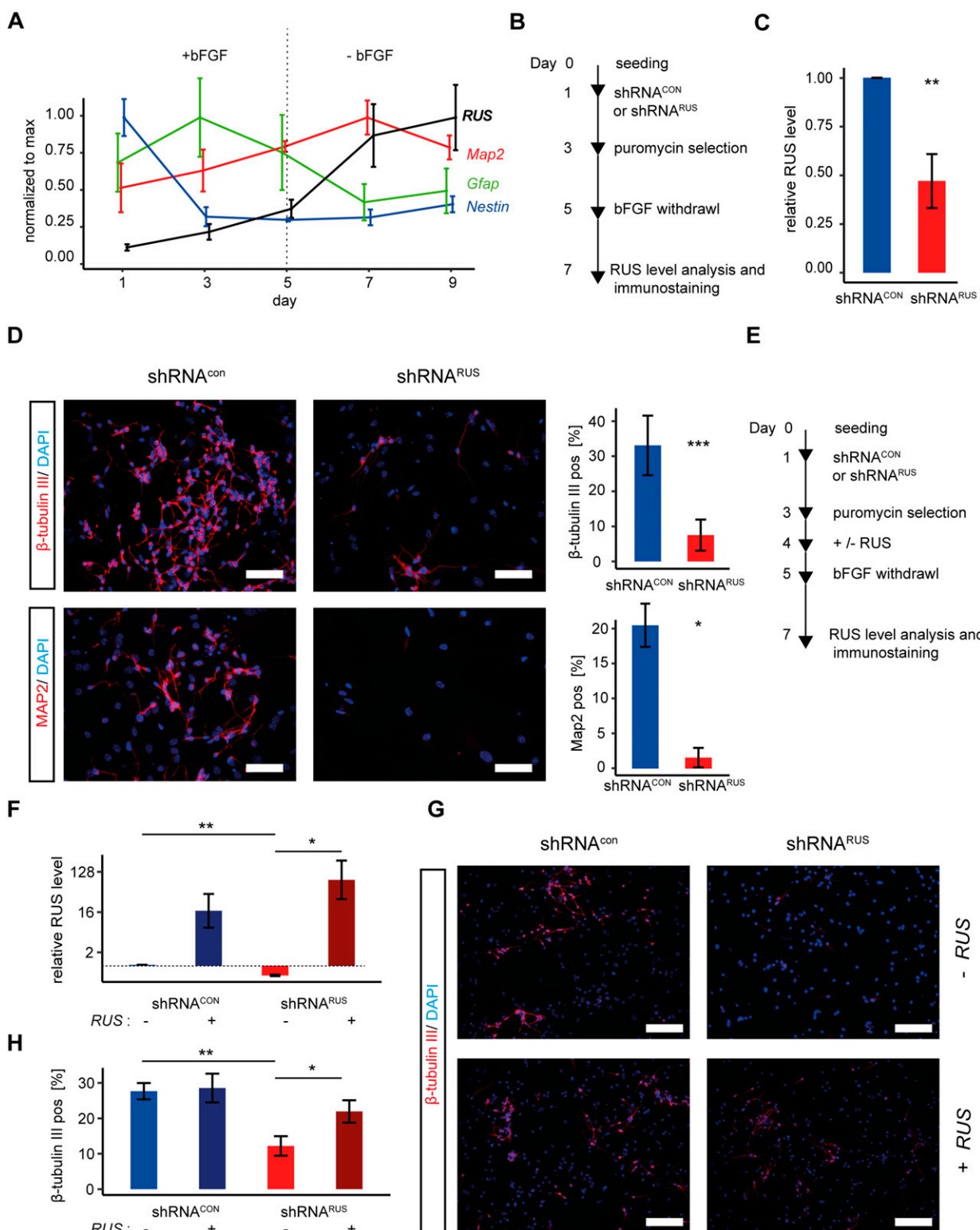

**Figure 2.** *RUS* **is involved in neuronal differentiation of murine embryonic cortical neural stem cells (NSCs).**
**(A)** RT–qPCR analysis of expression of *RUS*, *Map2*, *Gfap*, and *Nestin* transcripts as indicated, during a 9-d time course of murine embryonic cortical NSC differentiation. Values were normalized to the maximal expression of each RNA during the time course. Error bars show the standard error of the mean of three independent experiments. bFGF: basic Fibroblast growth factor. **(B)** Experimental strategy to deplete *RUS* in differentiating NSC by expressing shRNAs upon lentiviral transduction. **(C)** *RUS* levels determined by RT–qPCR in *RUS* knockdown cells (red, expressing shRNA[RUS]) compared with control cells (blue, expressing a scrambled shRNA[CON]). Error bars show the SD of the mean of four individual experiments. **(D)** Immunofluorescence visualization (left) of *β*-tubulin III (upper panel) and Map2 (lower panel) in control

high expression of the neuronal markers Map2, Dcx, β-tubulin III, and Mapt (Figs 2A and S2A and B). The expression level of *RUS* continually increased along with the neuronal markers, reaching robust expression on day 5 of the differentiation process (Figs 2A and S2B).

To explore a potential involvement of *RUS* during neuronal differentiation, we depleted *RUS* by RNA interference, expressing a *RUS*-targeting shRNA (shRNA$^{RUS}$) upon lentiviral transduction into differentiating NSCs (Fig 2B and Table S2, [Moffat et al, 2006]). *The shRNA$^{RUS}$* was selected to have no predicted off-targets, whereas significantly reducing RUS levels. Upon expression of shRNA$^{RUS}$, *RUS* levels were typically reduced by ~50% compared with control cells expressing a scrambled control shRNA$^{CON}$ (Fig 2C). Remarkably, upon *RUS* depletion, the number of cells expressing the neuron-specific β-tubulin III or the dendritic marker Map2 were reduced to 37% and 8%, respectively (Fig 2D).

The specificity of the knockdown was assessed by a rescue experiment. *RUS*-depleted and control cells were transduced with lentiviruses expressing *RUS* isoform 1 driven by the strong CMV promoter (Fig 2E). RT-qPCR revealed that *RUS* was increased roughly 20-fold compared with endogenous, wild-type levels (Fig 2F). Immunostaining of the cells for β-tubulin III served as a proxy for neurogenesis (Fig 2G). *RUS* expression in cultures that had been depleted of endogenous RUS largely restored the number of β-tubulin III-positive cells but did not further increase this value in the presence of endogenous *RUS* (Fig 2H).

*RUS* depletion led to reduced cell numbers in culture, which may be a consequence of reduced cell proliferation or increased apoptosis. Our subsequent analysis suggested that both processes contribute to cell loss. To explore proliferation effects, we supplemented differentiating NSC cultures with BrdU and monitored its incorporation by immunostaining as a measure of replication (Fig S2C and D). *RUS* depletion reduced the number of BrdU-positive, proliferating cells by 93.7% (Fig S2D). We also probed for apoptosis. We replaced the puromycin resistance gene in the shRNA vector by a GFP gene to visualize knockdown cells while avoiding cell death because of puromycin selection (Fig S2C). Immunostaining for cleaved caspase 3 in GFP-positive cells revealed a ninefold increase in apoptosis in shRNA$^{RUS}$-expressing cells compared with a very low level in control cultures (Fig S2E). We conclude that the depletion of RUS in differentiating NSCs inhibits cell proliferation and induces apoptosis.

## Depletion of RUS locks neural progenitor cells in their differentiation stage

For an in-depth characterization of the shRNA$^{RUS}$ knockdown phenotype in differentiating NSC we monitored transcriptional changes by RNA-seq analysis. We established the transcriptome at days 5 and 7 after seeding, when endogenous RUS expression is drastically increased, in cells either treated with shRNA$^{RUS}$ or shRNA$^{CON}$ (Fig S3A and Table S3). RNA interference by shRNA$^{RUS}$ reduced *RUS* levels to roughly 50%, as before (Fig S3B). Despite this incomplete depletion, the principal component analysis of four replicates clearly separated shRNA$^{CON}$ and shRNA$^{RUS}$ transcriptome profiles at both time points (Fig S3C).

Next, we determined differentially expressed genes (Fig S3D and Table S3) and analyzed enriched gene ontology (GO) classifications (Mi et al, 2013) among the up- and down-regulated genes, separately for the two time points. Consistent with findings that many lncRNAs regulate the expression of genes in the vicinity of their sites of transcription, the expression of the Slitrk3 was significantly reduced after RUS depletion (Table S3). In addition, the depletion of *RUS* massively affected the transcriptome, arguing that *RUS* also acts *in trans*. On day 5, 4,978 genes (24%) were transcribed at elevated levels under reduced *RUS* levels and 4,586 genes (22%) were repressed (Fig S3D). The expression changes were even more profound on day 7, when 6,623 genes (30%) and 6,456 genes (29%) were up- or down-regulated, respectively.

In agreement with the observed increase in apoptosis upon *RUS* depletion, we found the GO annotations associated with "cell death" and "apoptosis" (represented by "positive regulation of apoptosis" in Fig 3A) enriched among the induced genes on both days 5 and 7, exemplified by genes encoding, Bak1, and Foxo3. Fig 3B shows these genes among the 50 most deregulated genes enriching for the GO annotations: "cell-death," "neurogenesis," "cell-cycle" and "microtubule-based process." Annotations represented by GO classifications "cell cycle" and "microtubule-based process" (Fig 3A) were most significantly enriched among the down-regulated genes on both days, in support of the reduced BrdU incorporation (Fig S2E) and indicative of proliferation arrest (Fig 3A and B). Interestingly, genes with GO annotations relating to "neurogenesis" and "neuron differentiation" were mildly enriched among the down-regulated on day 5, but strongly enriched among the induced genes on day 7 (Fig 3A and B). Of note, at this level of analysis direct and indirect effects cannot be distinguished.

To explore the effects of *RUS* depletion in our RNA-seq data in more detail, we determined the read counts of several prominent genes that characterize the in vitro differentiation process (Fig 3C). We assessed the proliferation state (*Pcna* and *Ki67*), the NSC/RGC markers *Sox2*, *Pax6*, and *Gfap* as well as the neuronal markers *Neurog2*, *Neurod1*, *Map2*, *Camk2a*, *Grin3a*, and *Gabrb1*. In addition, we focused on the Notch1/2 and sonic hedgehog (Shh) signaling pathways regulating the expansion of RGCs and transit-amplifying intermediate progenitor cell populations. *Notch1/2*, its ligand *Dll1* and their downstream effectors *Hes1*, *Neurog2*, and *Ascl1* form an oscillatory network that regulates RGC cell renewal (Hatakeyama & Kageyama, 2006; Wang et al, 2016; Ivanov, 2019; Sueda & Kageyama,

(shRNA$^{CON}$) and knockdown (shRNA$^{RUS}$) cells using specific antibodies (magenta). Nuclei were stained with DAPI (4',6-diamidin-2-phenylindol, blue). Scale bar = 25 μm. Quantification of percentage of immune-positive cells by ImageJ (right). The bar diagrams show the percentage of positive cells. Error bars show the SD of four independent experiments. **(E)** Experimental strategy to rescue the *RUS*-depletion phenotype in differentiating NSC by lentiviral overexpression of *RUS*. **(F)** *RUS* levels were determined by RT-qPCR in control (shRNA$^{CON}$) and knockdown (shRNA$^{RUS}$) cells. Where indicated (+), *RUS* was overexpressed from a CMV promoter. Error bars show the SD of four independent experiments. The dashed line highlights the level of *RUS* in (shRNA$^{RUS}$) cells. **(G)** β-tubulin III immunostaining in control (shRNA$^{CON}$) and knockdown (shRNA$^{RUS}$) cells as a function of *RUS* overexpression. Nuclei are stained with DAPI. Scale bar = 50 μm. **(G, H)** Quantification of β-tubulin-III immunostaining of cultures as in (G). Error bars show the SD of four independent experiments (*$P < 0.05$, **$P < 0.01$, ***$P < 0.005$).

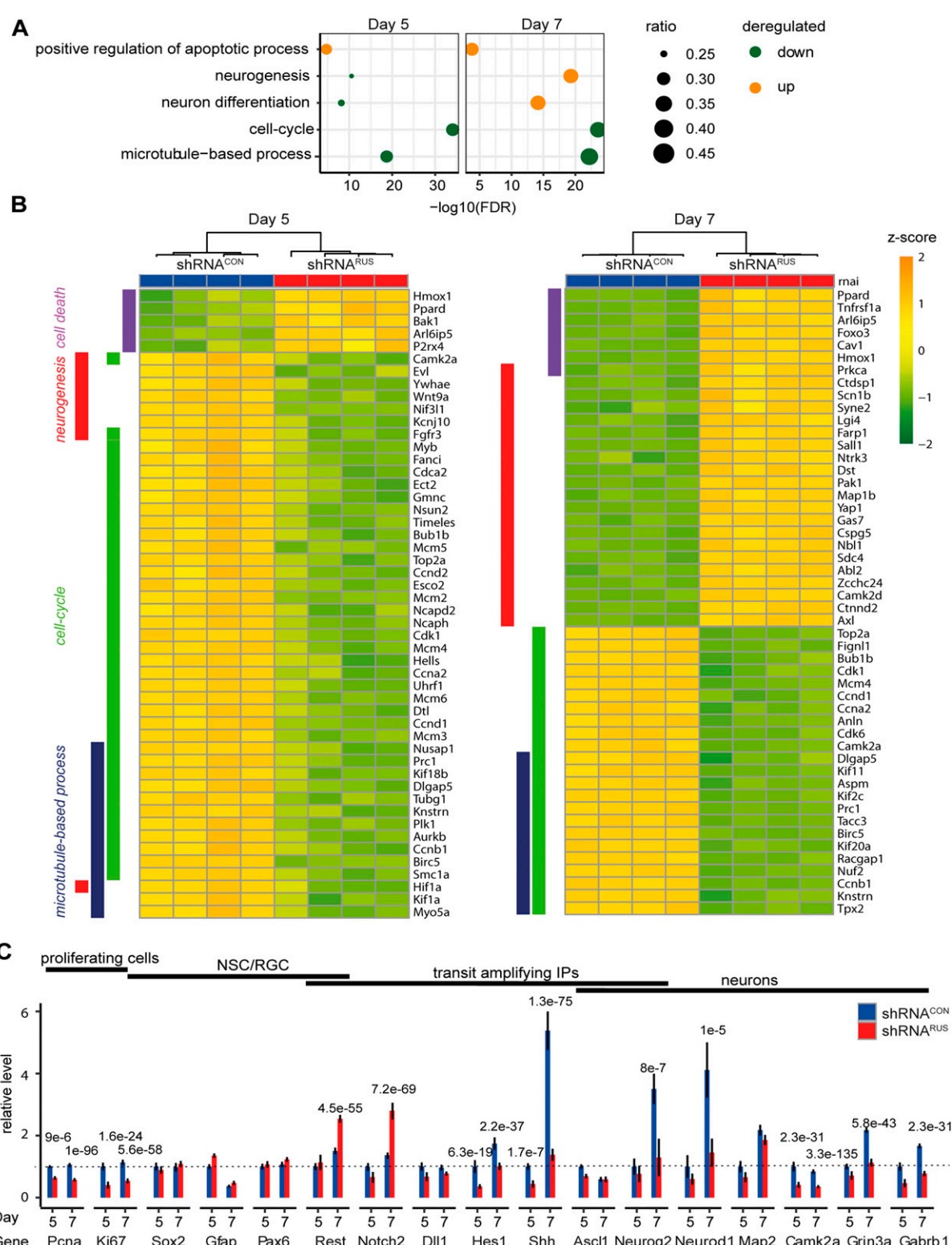

**Figure 3. Transcriptome changes upon depletion of *RUS*.**

**(A)** Enriched gene ontology (GO) classifications among genes down-regulated (blue) or up-regulated (orange) upon *RUS* depletion at days 5 and day 7 of culture, as indicated. Circle size indicates the number of deregulated genes compared with the total number of genes enriched in the respective GO annotation (100% = 1). **(B)** Heat map showing the top 50 deregulated genes enriching for the GO annotations "cell-death," "neurogenesis," "cell-cycle," and "microtubule-based process" on day 5 (left) and day 7 (right) of culture. Note that these are different genes. The genes were sorted by GO annotations and difference between shRNA$^{CON}$ and shRNA$^{RUS}$. **(C)** Expression levels of the indicated marker genes on day 5 and day 7 of culture in control (shRNA$^{CON}$, blue) and knockdown (shRNA$^{RUS}$, red) cells were determined by RNA-seq (TPM values were normalized to those of the control cells on day 5. Error bars show the SD).

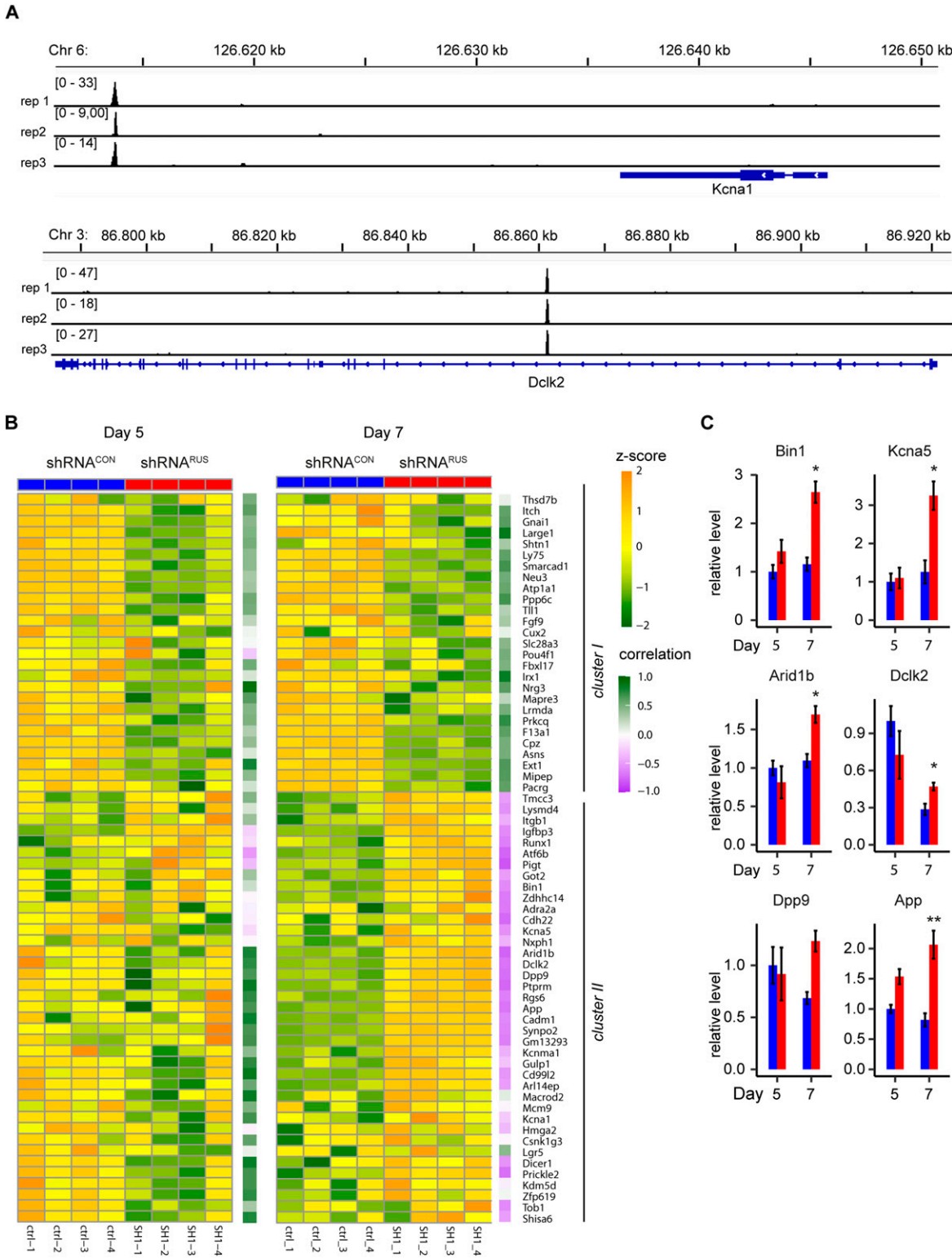

**Figure 4. Localization of *RUS* to chromosomal sites.**
**(A)** Browser view of two examples of *RUS* localization close to relevant neurogenic genes. The *RUS* ChIRP tag density of the three replicates is plotted in separate tracks in the genomic regions of the *Kcna1* (top) and *Dclk2* (bottom) genes. For orientation, the respective chromosomal regions are displayed above and the gene models below the traces. **(B)** Heat map showing the expression changes of 66 *RUS* putative target genes upon *RUS* depletion (shRNA^RUS, red) or in control cells (shRNA^CON, blue) on days 5 and 7. Replicate identifiers are indicated below the columns. Genes were hierarchically clustered using Euclidean distance based on their combined expression on both days. This yields two clusters depending on whether genes are activated or repressed upon *RUS* depletion. The gene names are indicated to the right of the 7-d heat map.

2019). We also included *Rest* as a transcriptional repressor of neuro-specific genes which helps to maintain the NSC state (Schoenherr & Anderson, 1995; Mukherjee et al, 2016).

Our RNA-seq analysis confirmed that the proliferative markers *Pcna* and *Ki67* were robustly down-regulated on both day 5 and day 7 (Fig 3C). The NSC/RGC markers *Sox2*, *Pax6*, and *Gfap* were less affected. However, the substantially reduced expression of the neuronal cell fate commitment markers *Hes1*, and *Shh* as well as of the neuronal markers: *Neurog2*, *Neurod1*, *Camk2a*, *Grin3a*, and *Gabrb1* confirmed our earlier notion that depletion of *RUS* compromises neuronal differentiation. Of note, the expression of those genes that are most strongly induced during neurogenesis between days 5–7 (i.e., *Shh*, *Neurog2*, and *Neurod*) was most strongly affected by *RUS* depletion (Fig 3C). The increased expression of *Notch2* is consistent with the observed maintenance of NSC/RGC markers, the reduced expression of cell cycle genes as well as genes involved in neuro-genesis (Engler et al, 2018; Mase et al, 2021). The induction of *Rest* at day 7 suggests a mechanism involving chromatin regulation.

We conclude that *RUS* is required for efficient proliferation and for differentiation of neuronal precursor cells in this in vitro system. The concomitant inhibition of cell proliferation (and hence cell renewal) and neurogenic differentiation may leave neuronal pro-genitor cells with conflicting signals that trigger apoptosis. The observation that at day 7 the most deregulated genes with an-notated GO term "neurogenesis" are activated upon *RUS* depletion (Fig 3B) prompts the speculation that *RUS* may be involved in the repression of transcription. Again, direct and indirect effects cannot be distinguished at this point.

## RUS associates with chromatin of key neurodevelopmental genes

As a first step towards defining the mechanism through which *RUS* regulates gene expression, we determined the subcellular locali-zation of *RUS*. After 2 d in culture, cells were fractionated into the cytoplasm, nucleoplasm and chromatin. RT-qPCR analyses showed that *RUS* is enriched in the chromatin fraction, similar to the splicing-associated lncRNA *MALAT* (Fig S4A).

To explore whether *RUS* localizes to specific chromosomal re-gions like other regulatory lncRNAs, we applied the ChIRP (Chro-matin Isolation by RNA Purification) methodology (Chu et al, 2011). Cells were harvested at day 7 of differentiation and *RUS* was iso-lated by hybridization with two independent probe sets ("odd" and "even"). The experiment was carried out in biological triplicate. All three isolations effectively retrieved *RUS* (~30% of input) and strongly enriched *RUS* over control RNAs *TBP mRNA*, *MALAT*, and *XIST* (Fig S4B). Between 157 to 203 peaks were scored in individual experiments, of which 129 (67%, Fig S4C) overlapped in all three experiments (Table S4).

Although we considered only peaks enriched by both probe sets, several enriched genomic sites contained sequences with similarity to one of the used oligonucleotide probe sequences. After re-moving them, 94 high-confidence putative *RUS* binding sites

remained for further analysis (for simplicity called "*RUS* binding sites" below). Genomic annotation revealed that four of them (4.3%) mapped to promoters, but the majority predominantly localized to intergenic (35.1%) or intronic (28.7%) regions, compatible with long-range regulatory elements. About a third of the locations mapped close to degenerate repetitive elements of various types, such as LINEs (4.2%), SINEs (12.8%), LTR (6.4%), and simple repeats (8.5%) (Fig S4D). GO analysis of the active genes next to *RUS* binding sites yielded an enrichment of the terms "forebrain development," "neurogenesis," and "generation of neurons." Among those are the genes encoding the microtubule-stabilizing protein Dclk2 and the potassium voltage-gated channel Kcna1 (Fig 4A, two further tracks: *Arid1b* and *Bin1* in Fig S4E). Both genes play a pivotal role in neuron differentiation (Shin et al, 2013; Chou et al, 2021).

Following the hypothesis that *RUS* binding to chromatin is in-volved in regulating near-by genes, we determined the expression changes of genes residing next to *RUS* binding sites (referred to as "putative target genes" henceforth) using the RNA-seq data of *RUS* knockdown samples. Of the 94 putative target genes, 66 were ro-bustly expressed in differentiating NSC (Fig 4B). The number of genes that changed their expression increased from day 5 to day 7 (54% and 77% of genes with altered expression, respectively), in line with the increase of *RUS* expression between days 5 and 7 of differentiation (Table S4).

Hierarchical clustering of expression separates putative target genes into two distinct clusters (Fig 4B). Cluster I contains genes significantly down-regulated on both days, whereas cluster II represents genes with enhanced expression, predominantly on day 7. The heat map shows several cluster II genes with reduced ex-pression on day 5 after *RUS* depletion. Because *RUS* depletion was less effective on day 5, we calculated the overall correlation of *RUS* expression and its putative target genes (Fig 4B, purple-to-green boxes to the right of heat maps). If we assume direct effects of *RUS* binding on target gene expression, we expect a positive correlation of genes with reduced expression with *RUS* depletion (essentially genes in cluster I) and a negative correlation of genes with en-hanced expression upon *RUS* depletion (predominantly cluster II genes on day 7). This is indeed largely the case (Fig 4B). Quanti-fication of the mRNA levels of *Bin1*, *Kcna5*, *Arid1b*, *Dclk2*, *Dpp9*, and *App* by RT-qPCR confirmed the increase in these target genes after *RUS* depletion on day 7 (Fig 4C). Remarkably, the expression of genes that are repressed on day 5 and activated on day 7, for example, *Arid1b*, *App*, and *Kcna1* (Fig S4F), correlates positively on day 5 and negatively on day 7 with *RUS* expression, in support of a direct effect of *RUS* on close-by genes. Our results thus suggest that *RUS* may mediate both, activating and repressive regulation.

## RUS interactors suggest epigenetic regulatory mechanisms

LncRNAs usually elicit their gene regulatory effects through interacting effector proteins. To explore how *RUS* may mediate both, activating and repressive functions, we sought to identify

The purple-green code to the right of each individual heat map indicates the degree of correlation between *RUS* and putative target gene expression. **(C)** Expression levels of the putative target genes: Bin1 (n = 3 or 7 [3/7] for days 5 or 7, respectively), Kcna5 (n = 7/7), Arid1b (n = 3/9), Dclk2 (n = 3/4), Dpp9 (n = 5/5), and App (n = 3/5) on day 5 and day 7 of culture in control (shRNA^CON, blue) and knockdown (shRNA^RUS, red) cells were determined by RT-qPCR (values were normalized to those of the control cells on day 5, error bar show the standard error of the mean, *P < 0.05, **P < 0.01, ***P < 0.005).

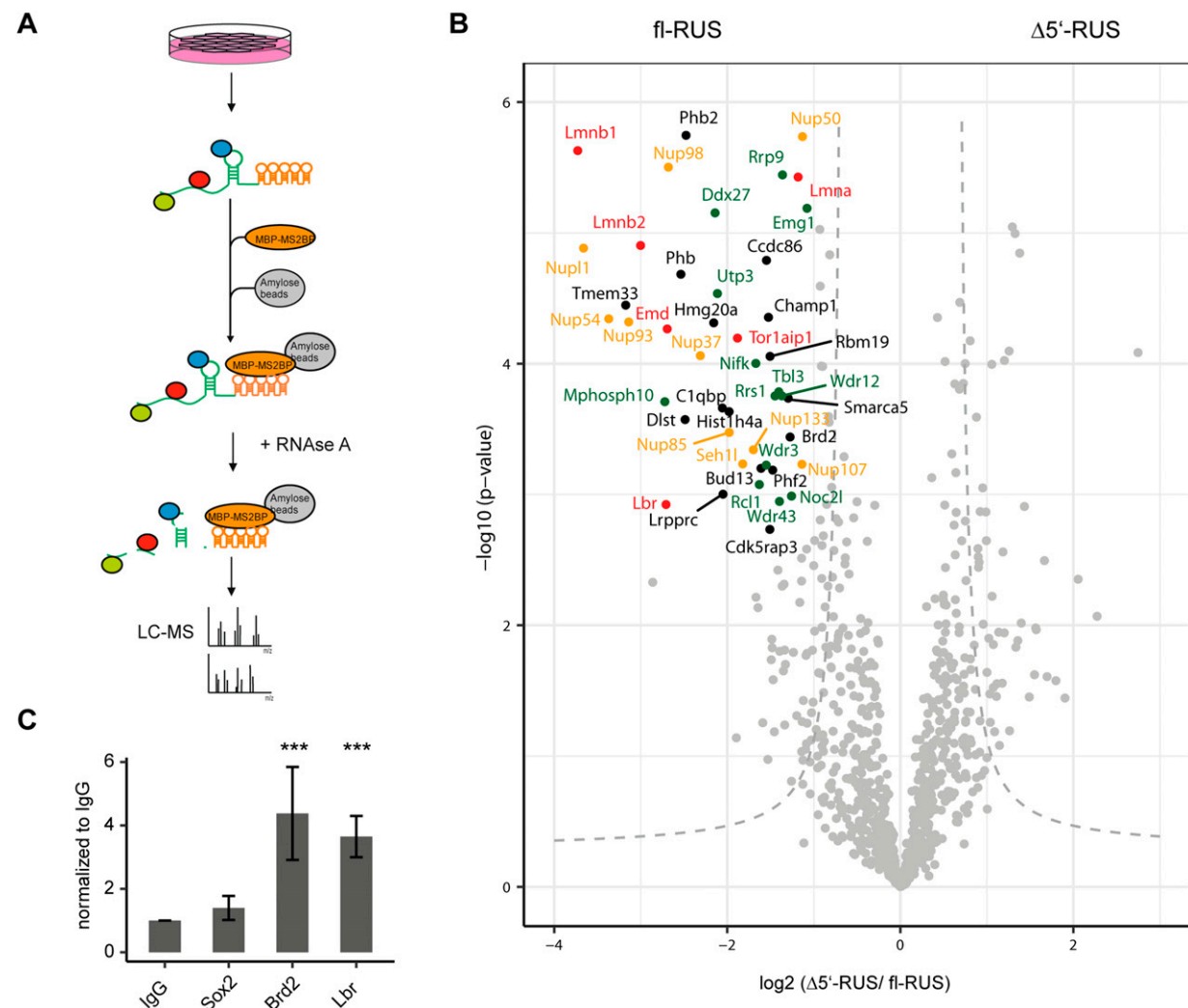

**Figure 5. RUS interacts with components of the nuclear pore, -lamina, and nucleolus.**
**(A)** Schematic overview of the affinity purification of *RUS*-interacting proteins (colored spheres). *RUS* RNA (green), tagged with five MS2 stem-loop structures (orange) is stably expressed in Neuro2A cells. The RNA is affinity-purified by binding to MS2BP-maltose binding protein on an amylose resin. For details, see text. **(B)** Volcano plot showing affinity-purified nuclear proteins that bind differentially to full-length *RUS* (left) or a *RUS* RNA from which exon 1 was deleted (Δ5'-*RUS*). Proteins with a change greater than 2 and a *P*-value smaller than 0.002 are considered robust interactors and annotated by their gene name. The dashed gray hyperbolic curves depict a permutation-based false discovery rate estimation (*P* = 0.05; s0 = 1). Some proteins are color-coded: proteins of the nuclear lamina (red), nuclear porins (orange), and nucleolar proteins (green). **(C)** RT-qPCR analysis of *RUS* co-immunoprecipitated with antibodies against Sox2, Brd2, Lbr, and control IgG from differentiating neural stem cells. Error bars show the SD (***P* < 0.01 compared with IgG purification).

*RUS*-binding proteins. When mouse and human *RUS* sequences are compared, a remarkable degree of conservation of exon 1 stands out (Fig 1C). Because such conservation may be indicative of important functional interactions, we compared interactors of complete *RUS* with a 5'-deleted RNA (Δ5'-*RUS*), lacking exon 1. Both RNAs were tagged with 5 MS2 stem-loop structures at the 3' end, enabling affinity purification via binding to MS2-binding protein (MS2BP) (Johansson et al, 1997; Zhou et al, 2002; Tsai et al, 2011).

Because differentiating NSCs cannot be obtained in sufficient amounts for RNA-affinity purification, we established an RNA-affinity purification protocol using the well-established Neuro2A cell line. *RUS* is normally not expressed in these cells and so our experiment identifies potential protein interactors that are not

relevant in these cells. To assure an equivalent expression of both RNAs, we first generated Neuro2A derivatives by inserting an FRT recombinase site into the genome through lentiviral transduction. These clonal cells were then transfected with FRT-flanked *RUS* expression constructs along with a flipase expression plasmid (Andrews et al, 1985; Sauer, 1994; See et al, 2002). Clones containing integrated *RUS* expression cassettes were expanded and analyzed. These clones express comparable levels of either full-length *RUS* or Δ5'-*RUS*.

Lysates of *RUS*- and Δ5'-*RUS*-expressing cells were incubated with recombinant MS2-binding protein (MS2BP), which in turn was tagged with a maltose-binding protein (MBP) (see scheme in Fig 5A). MS2BP-bound RNA was retrieved by absorption of MBP to amylose beads, captured proteins were eluted with RNAse A treatment and

Table 1.   Table includes affinity-purified nuclear proteins that bind more than full length *RUS* (*P*-value < 0.002, $\log_2$(mut/fl RUS) < −1) and the localization to nuclear compartments as nucleolus, nuclear lamin, and nuclear pore.

| UniProt ID | Gene name | $-\text{Log}_{10}$ (*P*-value) | $\text{Log}_2$(mut/fl) | Only detected by full length RUS | Nuclear compartment |
|---|---|---|---|---|---|
| Q7JJ13 | Brd2 | 3.44 | −1.27 | No | — |
| Q8R149 | Bud13 | 3.20 | −1.61 | No | — |
| O35658 | C1qbp | 3.66 | −2.06 | No | — |
| Q9JJ89 | Ccdc86 | 4.79 | −1.55 | No | — |
| Q99LM2 | Cdk5rap3 | 2.73 | −1.51 | No | — |
| Q8K327 | Champ1 | 4.35 | −1.52 | No | — |
| Q921N6 | Ddx27 | 5.15 | −2.14 | No | Nucleolus |
| O08749 | Dld | 6.00 | −2.13 | No | — |
| Q9D2G2 | Dlst | 3.57 | −2.49 | No | — |
| O08579 | Emd | 4.27 | −2.69 | No | Nuclear lamin |
| O35130 | Emg1 | 5.19 | −1.08 | No | Nucleolus |
| P62806 | Hist1h4a | 3.63 | −1.98 | No | — |
| Q9DC33 | Hmg20a | 4.31 | −2.16 | No | — |
| P38647 | Hspa9 | 6.57 | −1.99 | No | — |
| Q3U9G9 | Lbr | 2.92 | −2.71 | No | Nuclear lamin |
| P48678 | Lmna | 5.43 | −1.18 | No | Nuclear lamin |
| P14733 | Lmnb1 | 5.63 | −3.73 | No | Nuclear lamin |
| P21619 | Lmnb2 | 4.90 | −3.00 | No | Nuclear lamin |
| Q6PB66 | Lrpprc | 3.00 | −2.05 | No | — |
| Q810V0 | Mphosph10 | 3.71 | −2.72 | No | Nucleolus |
| Q91VE6 | Nifk | 4.00 | −1.67 | No | Nucleolus |
| Q9WV70 | Noc2l | 2.99 | −1.26 | No | Nucleolus |
| Q8BH74 | Nup107 | 3.23 | −1.14 | No | Nuclear pore |
| Q8R0G9 | Nup133 | 3.34 | −1.70 | No | Nuclear pore |
| Q9CWU9 | Nup37 | 4.06 | −2.31 | No | Nuclear pore |
| P59235 | Nup43 | 6.28 | −2.93 | No | Nuclear pore |
| Q9JIH2 | Nup50 | 5.74 | −1.13 | No | Nuclear pore |
| Q8BTS4 | Nup54 | 4.34 | −3.37 | No | Nuclear pore |
| Q8R480 | Nup85 | 3.47 | −1.98 | No | Nuclear pore |
| Q8BJ71 | Nup93 | 4.32 | −3.14 | No | Nuclear pore |
| Q6PFD9 | Nup98 | 5.50 | −2.68 | No | Nuclear pore |
| Q8R332 | Nupl1 | 4.88 | −3.66 | No | Nuclear pore |
| P67778 | Phb | 4.68 | −2.54 | Yes | — |
| O35129 | Phb2 | 5.75 | −2.48 | Yes | — |
| Q9WTU0 | Phf2 | 3.19 | −1.48 | No | — |
| Q8R3C6 | Rbm19 | 4.06 | −1.51 | No | — |
| Q9JJT0 | Rcl1 | 3.08 | −1.63 | No | Nucleolus |
| Q91WM3 | Rrp9 | 5.44 | −1.36 | No | Nucleolus |
| Q9CYH6 | Rrs1 | 3.75 | −1.45 | No | Nucleolus |
| Q8R2U0 | Seh1l | 3.23 | −1.82 | No | Nuclear pore |
| Q91ZW3 | Smarca5 | 3.74 | −1.30 | No | — |
| Q8C4J7 | Tbl3 | 3.78 | −1.41 | No | Nucleolus |

**Table 1.  Continued**

| UniProt ID | Gene name | −Log$_{10}$ (*P*-value) | Log$_2$(mut/fl) | Only detected by full length *RUS* | Nuclear compartment |
|---|---|---|---|---|---|
| Q9CR67 | Tmem33 | 4.45 | −3.17 | No | — |
| Q921T2 | Tor1aip1 | 4.20 | −1.88 | Yes | Nuclear lamin |
| Q9JI13 | Utp3 | 4.54 | −2.11 | Yes | Nucleolus |
| Q9JJA4 | Wdr12 | 3.75 | −1.37 | No | Nucleolus |
| Q8BHB4 | Wdr3 | 3.22 | −1.55 | No | Nucleolus |
| Q6ZQL4 | Wdr43 | 2.95 | −1.40 | No | Nucleolus |

Table highlights whether a protein was identified by full-length *RUS* only.

identified by LC–MS, using label-free quantification (LFQ) (Cox & Mann, 2009).

Full-length *RUS* enriched many more proteins in comparison to Δ5′-*RUS* (Fig 5B and Table S5). While we cannot exclude that this is due to the increased size of the *RUS* RNA, this seems unlikely given the size difference of 912 (*RUS*) versus 679 nucleotides (Δ5′-*RUS*). Proteins with a fold-change greater than 2 and a *P*-value smaller than 0.002 were considered robust and specific binders. Only nine proteins were purified selectively along with Δ5′-*RUS*. By contrast, 49 proteins were enriched by co-purification with the full-length construct and therefore considered exon 1-specific interactors (Tables 1 and S5). Among them, Phb, Phb2, Tor1aip1, and Utp3 were purified exclusively by the full-length *RUS* RNA.

Phb and Phb2 correspond to the prohibitin complex, a mito-chondrial regulator with neuroprotective functions and nuclear co-repressor of cell cycle-regulated genes (Koushyar et al, 2015).

We also found find numerous components of the nuclear periphery, most prominently subunits of the nuclear pore complex (Nupl1, Nup37, Nup43, Nup50, Nup54, Nup85 Nup93, Nup98, Nup107, Nup133, and Seh1l orange in Fig 5B) and several constituents of the nuclear lamina: emerin (Emd), lamins A, B1, and B2 (Lmna, Lmnb1, and Lmnb2), lamin B receptor (Lbr) as well as the lamin A/B binding protein Tor1aip1 (red in Fig 5B).

Furthermore, *RUS* exon 1 retrieved many nucleolar proteins (Ddx27, Emg1, Mphosph10, Noc2l, Nifk, Rcl1, Rrp9, Rrs1, Tbl3, Utp3, Wdr3, Wdr12, and Wdr43 green in Fig 5B) and some interesting chromatin regulators (e.g., the bromodomain protein Brd2, the chromatin constituent Hmg20a, the nucleosome remodeling ATPase Smarca5, the lysine demethylase subunit Phf2, and the RNA helicase Ddx54).

The finding of robust interaction of *RUS* with nuclear pores and the lamina suggest well-established epigenetic regulatory mechanisms (to be discussed below). Binding of lncRNA *Xist* to Lbr has been suggested to tether the inactive X chromosome to the nuclear envelope, which forms a silent compartment (Chun-Kan et al, 2016). To validate the binding between *RUS* and Lbr, we returned to our NSC differentiation model. Nuclear extracts were prepared from cells harvested at day 7 of differentiation. Lbr was immunoprecipitated and co-precipitated RNA quantified by RT-qPCR. *RUS* was retrieved 3.7-fold more by comparison to an anti-IgG purification (Fig 5C). Parallel reactions confirmed the selective interaction of Brd2 with *RUS*, whereas Sox2 served as a control.

In summary, our data support the idea of the long, noncoding RNA *RUS* as a crucial regulator of the neurogenic gene expression program through epigenetic mechanisms.

# Discussion

### The lncRNA RUS is required to execute the neurogenic program

Our study presents a first functional characterization of the lncRNA *LINC01322*, which we term *RUS* (for RNA upstream of *Slitrk3*). Like other neurogenic lncRNAs, *RUS* is well conserved in mammals by sequence and synteny next to the neurodevelopmental gene *Slitrk3*. It is predominantly expressed in neural tissues. Although the RNA bears some coding potential, we did not detect any of the theoretically encoded peptides. *RUS* associates with chromatin at specific sites in the vicinity of neurodevelopmental genes and interacts with several proteins involved in epigenetic gene regulation, suggesting that *RUS* acts as lncRNA. However, at this point we cannot exclude the formal possibility that a fraction of *RUS* is processed to functionally relevant miRNAs.

Transcriptome analyses revealed that sh-mediated depletion of *RUS* results in massive gene expression changes. In fact, approximately half of all genes were affected to a certain degree. The responses were equally divided between gene activation and repression and were modulated during the 7 d of differentiation. This finding is interesting because most lncRNAs studied so far either mediate activation or repression (Rinn & Chang, 2020; Statello et al, 2021). Although indirect effects cannot be excluded yet, the fact that we found epigenetic activators and repressors bound to *RUS* exon 1 in pull-down experiments, supports the idea that *RUS* may mediate gene activation and repression in a highly context-dependent manner. Conceivably, *RUS* may function through diverse mechanisms, as emerges for the HOTAIR RNA (Price et al, 2021).

On day 5 of differentiation, reduced *RUS* levels correlate with reduced expression of many genes involved in neurogenesis and cell cycle, suggesting that the lncRNA promotes target gene expression to enable amplification of intermediate precursor cells and NSC differentiation. This is in line with the observation that *RUS* is expressed in hESC-derived LRGs (Ziller et al, 2015).

*RUS* is most highly expressed in the adult hippocampus, in which neurogenesis still occurs (Eriksson et al, 1998). Adult neurogenesis relies on expanding transit-amplifying IPs maintained by *Shh* expression (Antonelli et al, 2018) and differentiation by increased

*Neurog2* expression (Galichet et al, 2008). At day 7 of our differentiation time course, *Shh*, *Neurog2*, and *NeuroD1* are among the most repressed genes upon *RUS* depletion. In addition, we found a reduced expression of several subunits of glutamate and GABA receptors, such as *Grin3a* and *Gabrb1*, which are predominantly expressed in neurons.

Although the pattern of endogenous *RUS* expression and the observation that neuron formation was impaired after *RUS* depletion suggest a role of the lncRNA in promoting neuronal differentiation, RNA-seq and GO analysis revealed a significant up-regulation of neuronal differentiation genes on day 7 after *RUS* depletion. Such conflicting results may be a consequence of induction of proneuronal genes such as *Notch2* and *Rest* after *RUS* depletion. We speculate that *RUS* depletion locks neuronal precursors in an intermediate state towards neuronal differentiation, with arrested cell cycle. The activation of pro-apoptotic genes may result from perturbed cell identity. However, it is also possible that increased apoptosis after *RUS* depletion impaired neuron formation.

**Potential mechanisms of RUS-mediated gene regulation**

Given the diverse and presumably very site-specific effects of *RUS* function, we can only speculate about potential mechanisms. Our stringent ChIRP approach revealed a very consistent set of *RUS* interactions with a limited number of high-confidence chromatin loci. The localization of binding sites predominantly in introns and intergenic regions argue for long-range regulation. Considering that the RNA is not highly expressed, we speculate that its range of activity may be limited to the genes in the vicinity of tethering sites (Engreitz et al, 2016).

Remarkably, most of the genes closest to a *RUS* binding site were expressed in differentiating NSCs and changed their expression state upon *RUS* depletion. For example, *RUS* binds in the genome next to genes essential for cell cycle and neuronal differentiation, such as *Fgf9*, *Mapre3*, and *Ppp6c*, *Arid1b*, *Dclk2*, and *Kcna1*. The expression of these critical genes is affected by *RUS* depletion. Furthermore, *RUS* binding sites can be observed in introns of the E3 ubiquitin ligase genes *Itch* and *Fbxl17*. Itch ubiquitinates Notch proteins for degradation to turn off Notch signaling (Chen et al, 2021). *Fbxl17* plays a pivotal role in Shh signaling by degrading Sufu to enable the translocation of Sufu-sequestered transcription factors to the nucleus (Raducu et al, 2016). Consequently, reduction of both factors after *RUS* depletion resulted in increased Notch signaling and reduced Shh signaling, consistent with our RNA-Seq data. Notch signaling is important for maintaining the active or quiescent NSC state by preventing neuronal differentiation (Sueda & Kageyama, 2019). Shh signaling regulates proliferation of neural precursors (Yao et al, 2016). By activating both genes *RUS* facilitates proliferation and ensures proper differentiation of neural precursor cells.

LncRNA often work by recruiting epigenetic regulators to locally concentrate them at target chromatin (Markaki et al, 2021). Our RNA-affinity purification relies on protein-*RUS* interactions formed under physiological conditions in intact cells and purifying complexes under native conditions. Because we wished to identify proteins interacting with the conserved exon 1 of *RUS*, we monitored the differential binding to RNA containing or lacking this sequence. This is a stringent approach because functionally meaningful proteins may well (and are indeed likely to) bind to the remainder of *RUS* as well, but they are not discussed here (but see Table S5). In the following, we discuss hypothetical scenarios, in which *RUS* recruits regulatory functions to chromosomal target loci. It is also possible that *RUS* sequesters the factors in competition with other interactors, which would have opposite effects on gene regulation compared with recruitment scenarios (Xi et al, 2022).

Among the proteins purified by full length *RUS* only, the prohibitin complex (consisting of Phb and Phb2) stands out. Prohibitin has functions in several cellular compartments, including mitochondria and nuclei (Wang et al, 2002; Fusaro et al, 2003; Rajalingam & Rudel, 2005; Koushyar et al, 2015). Prohibitin has been termed an oncogene, as it promotes proliferation and dedifferentiation in neuroblast cells (MacArthur et al, 2019) and a tumour suppressor gene beacuse it was shown to inhibit the cell cycle by repressing E2F-regulated genes via recruitment of the retinoblastoma protein and histone deacetylases (Wang et al, 2002). It is tempting to speculate that tethering the Phb complex to chromatin contributes to inhibition of proliferation and activation of apoptosis.

Strikingly, the RNA pull-down retrieved numerous proteins of the nuclear envelope. We scored six constituents of the nuclear lamina, including three types of lamins and lamin B receptor (Lbr). The inner nuclear membrane assembles a well-known repressive compartment to which inactive heterochromatin is tethered. These lamina-associated domains may be constitutive or facultative (van Steensel & Belmont, 2017). Conceivably, *RUS* mediates tethering of genes destined to be silenced to the lamina, where they acquire heterochromatic features. Such a scenario has precedent in the finding that the lncRNA *XIST* promotes X chromosome inactivation in female cells by tethering the target chromosome to the nuclear envelope via Lbr (Chun-Kan et al, 2016).

Repressive heterochromatin is also found at the surface of nucleoli (Kind et al, 2013; Vertii et al, 2019). Remarkably, we found 13 nucleolar proteins enriched specifically by *RUS* exon 1, which further supports the speculation that *RUS* partitions genes into silencing compartments. However, some of the retrieved nucleolar proteins also have nuclear functions. For example, NOC2L (NOC2 Like Nucleolar Associated Transcriptional Repressor, also known as NIR) associates with p53 in the nucleus to repress a subset of p53-target genes, including p21, by inhibition of histone acetylation (Hublitz et al, 2005). Interestingly, the exon 1 interactor NIFK (also a nucleolar protein with nuclear functions) also cooperates with p53 to silence the p21 promoter during checkpoint control (Takagi et al, 2001). Apparently, *RUS* also contributes to p21 silencing because the gene gained activity upon depletion of the lncRNA. Similarly, the exon-1 interactor Cdk5rap3 activates p53 activity by repressing its degradation by Hdm2 (Wang et al, 2006). Such a scenario provides a plausible and testable hypothesis for the observed cell cycle arrest at reduced *RUS* levels.

In addition to constituents of the nuclear lamina, we found 11 nuclear pore components (Nup11, Nup37, Nup43, Nup50, Nup54, Nup85, Nup93, Nup98, Nup107, Nup133, and Seh1l) among the exon 1 interactors. In addition to nuclear transport, the nuclear pore complex plays an important role in transcriptional regulation and cell identity, apparently by generating a microenvironment that fosters epigenetic regulation of associated genes (Pascual-Garcia & Capelson, 2021). In *Drosophila*, Nup93 is associated with genes

repressed by the polycomb complex and is required for efficient repression (Gozalo et al, 2020).

By contrast, three nucleoporins bound RUS are predominantly associated with transcriptional activation. Nup98 acts as anchor point for enhancer (Pascual-Garcia et al, 2017) and activates transcription by recruiting the Wdr82-Set1A/COMPASS complex to regulate H3K4 trimethylation (Franks et al, 2017). Similarly, Nup107 and Seh1l activate transcription by assembling transcription factor (TF) complexes at the nuclear pore (Liu et al, 2019). It is tempting to speculate that RUS may mediate facultative association of gene loci with the nuclear periphery, which would then be subject to regulation of the corresponding microenvironment. This may initially involve an initial transcriptional activation to execute the differentiation programme. The subsequent compartmentalization of chromosomal loci into a repressive environment may serve to terminally silence cell cycle genes in mature neurons.

The exon 1 interactor HMG20A (also known as iBraf) is known to antagonize repressive LSD1–REST complexes. Because LSD1–REST–dependent H3K4 demethylation represses neuronal genes, HMG20A action promotes neuronal differentiation (Ceballos-Chávez et al, 2012; Garay et al, 2016). The interaction of RUS with HMG20A, therefore, likely affects neuronal differentiation, but whether the outcome is positive (through recruitment) or negative (through squelching) remains to be explored. Of note, REST expression increases upon RUS depletion, consistent with the observed inhibition of neurogenesis.

In summary, our mapping of putative target genes and RUS interactors are compatible with a range of testable, hypothetical and not mutually exclusive scenarios that may explain the observed change in phenotype and gene expression upon RUS depletion during differentiation of NSCs. We propose that RUS may be involved in several aspects of the neurogenic program in a highly context-dependent manner, including amplification of precursor cells and terminal neuronal differentiation.

# Materials and Methods

Used reagents, tools, and oligonucleotides are listed in Tables S1 and S2.

### Cultivation and differentiation of primary NSCs

The isolation of cortical embryonic stem cells from E15-E16 murine cortices was approved by the animal welfare committees of LMU and the Bavarian state. Cortices were dissected from pooled mixed-sex embryonic brains, washed five times with Hanks Balanced Salt Solution and incubated in 0.5% trypsin–EDTA for 15 min. Cortices were then washed five times with MEM-HS supplemented with L-glutamine, essential amino acids, nonessential amino acids, and 10% horse serum. The single cells in suspension were pelleted at 200$g$ for 5 min, and seeded at a density of $5 \times 10^5$ cells/ml. NSCs were cultured in DMEM-F12 with 5% FCS, B27 supplement and 20 ng/ml basic fibroblast growth factor (bFGF) on poly-D-lysine-coated culture dishes at 37°C in 5% $CO_2$ (Kilpatrick & Bartlett, 1993; Johe et al, 1996; Azari et al, 2011; Mukhtar et al, 2020). Every second day,

the culture medium was supplemented with 20 ng/ml bFGF. Cells were passaged up to six times by trypsin digestion at 95% confluency by 1:2 dilution. Differentiation was induced 5 d after seeding in neurobasal medium with B27 supplement/0.25× GlutaMAX.

For RT-qPCR analysis or RNA-seq experiments, $3 \times 10^5$ NSCs were seeded in 2 ml medium on 35-mm dishes. For microscopy experiments, $1.6 \times 10^5$ NSCs were seeded in 1 ml medium on 12.8-mm dishes equipped with 12-mm coverslips.

Sh-mediated knockdown experiments were started 1 d after seeding by addition of 5 $\mu$l virus per 35-mm dish or 3 $\mu$l KD virus per 12.8-mm dish. To restore RUS expression, 10 or 6 $\mu$l RUS overexpression-virus per 35 or 12.8 mm dish, respectively, was added to KD cells 4 d after seeding.

### Cultivation of Neuro2A cells

Neuro2A cells were cultured in DMEM-GlutaMAX and 10% FCS at 37°C in 5% $CO_2$.

### Immunohistochemistry

Cells were plated on poly-L-lysine-coated glass plates in a 24-well plate. All cell washes were carried out in PBS, all incubations were at RT. Cells were fixed in 4% PFA for 20 min at RT, washed once for 10 min, and blocked with blocking solution (0.3% Triton X-100, 2% donkey serum in PBS) for 30 min. The primary antibody (1:1,000) was diluted in 200 $\mu$l blocking solution and added for 1 h 30 min while shaking. The antibody solution was removed, and the cells were washed three times for 10 min. Cells were incubated with the secondary antibody (1:2,000) in 200 $\mu$l blocking solution for 1 h 30 min as before. After three 10-min washes, nuclei were stained for 15 min using DAPI (2-[4-amidinophenyl]-6-indolecarbamidine dihydrochloride) 1:1,000 in PBS. The cells were mounted in the presence of diazabicyclo-octane (DABCO). Stained cells were analyzed with a Leica DM8000 fluorescent microscope, and images were quantitatively processed with ImageJ. Images from DAPI and antibody staining were thresholded, colocalized, and watershed-transformed. The particles in the resulting overlay image were counted using the particle analyzer. Per experiment, 3–5 microscope fields on 3–4 plates each were recorded and analyzed.

### BrdU labeling

Cull culture medium was supplemented with 1 $\mu$g/ml bromodesoxyuridine. After 24 h, cells were immunostained with an anti-BrdU antibody.

### Quantitative reverse transcription-PCR (RT-qPCR)

RNA from cells, tissues or biochemical experiments was extracted with Trizol and chloroform and precipitated using 50% isopropanol and 15 $\mu$g linear acrylamide. RNA was washed twice with 75% EtOH, dissolved in nuclease-free water, and reverse-transcribed using MMuLV RT (Thermo Fisher Scientific) and oligodT(18-20). ChIRP and RIP-purified RNA was amplified with random hexamers. RT-qPCR analysis was performed with 1 $\mu$M of each primer in Fast SYBR Green Master Mix (Thermo Fisher Scientific). The ΔCt values were

normalized with amplicons detecting against TATA-binding protein (*TBP*) mRNA.

### 3′ RACE

The *RUS* 3′-end was cloned from a hippocampal RNA using the FirstChoice RLM-RACE Kit (Thermo Fisher Scientific). One microgram of RNA was reverse-transcribed using an anchored 3′ RACE oligo(dT) primer. This was followed by two rounds of nested PCR using RUS-3′-RACE as forward and 3′-outer primers and 3′-inner as reverse primer. The PCR product was gel-purified and sequenced.

### Generation of the RUS knockdown vector

ShRNAs were designed according to standard procedures (Yuan et al, 2004). In brief, 100 pmol *RUS*-sh-FW and 100 pmol RUS-sh-RV were annealed in 50 µl NEB2.1. The annealed fragment was cloned into pLKO.1-TRC-Puro vector, linearized with AgeI and EcoRI (Moffat et al, 2006), and amplified in Dh5α. For pLKO.1 vectors containing GFP as a selection marker, the puromycin resistance gene was replaced with the GFP gene via BamHI and KpnI restriction sites. Towards this end, the GFP cDNA was amplified from pLenti-CMV-GFP-Hygro (Campeau et al, 2009) by PCR using the primers: BamH-GFP-fw and Kpn-GFP-rv.

### Construction of pcDNA-5FRT-5xMS2

pcDNA.5-FRT vectors used to generate stable FlpIN Neuro2A cells were equipped with 5xMS2 stem-loops. The 3xMS2 stem-loop sequence was PCR-amplified with the primers MS2_fw and MS2_rv from pAdMl3-(MS2)$_3$, digested with BamHI and XbaI, and ligated to BamHI/ XbaI-linearized pcDNA5-FRT. Upon amplification in Dh5α, one clone fortuitously expanded 3xMS2 stem-loops to 5xMS2 stem-loops. This clone was used.

### Generation of RUS overexpression vector

*RUS* and Δ5′*RUS* sequences of isoform 1 missing exon 4 were isolated from a hippocampal cDNA library by PCR with the primers: RUS-LIC-fw or Δ5′ RUS-LIC-fw, respectively, and RUS-LIC-rv and cloned into pcDNA-5-FRT or pcDNA.5-FRT-5xMS2 (Thermo Fisher Scientific) via LIC cloning (Wang et al, 2012) and amplified in Dh5α. For rescue experiments, the *RUS* cDNA targeted by shRNA$^{RUS}$ was shuffled into pLenti-CMV-GFP-Hygro (Campeau et al, 2009) via ClaI and ApaI restriction sites to replace GFP and the hygromycin resistance gene. All constructs were verified by sequencing.

### Construction of pLenti-FRT

pLenti-GFP-Puro (Campeau et al, 2009) was digested with XbaI and BamHI to remove GFP downstream of the CMV promoter. FRT site was generated by annealing the oligonucleotides FRT_fw and FRT_rv. For annealing, 100 pmol of each oligonucleotide was heated in 50 µl NEB 2.1–95°C for 5 min and slowly cooled down. 2 µl annealing scale was ligated into 20 ng digested vector and transformed in Dh5α.

### Production of lentiviral particles

All lentiviral experiments were conducted according to standard protocols (Moffat et al, 2006) and approved by the Bavarian state. 3 × 10$^6$ HEK293T cells were seeded in 8 ml DMEM-GlutMax supplemented with 8% FCS on a 10 cm culture dish. Per virus production, four 10-cm dishes were seeded. Next day, 53 µg DNA in a molar ratio of 2:1:1 of lentiviral-vector: psPAX2: pMD2.G transfected into 50–70% confluent cells. The medium was changed next day. 2 d after transfection, viral particles were purified by sedimentation (87,000$g$, 2 h) from the medium and dissolved in 200 µl TBS5 (50 mM Tris–HCl, pH 7.8, 130 mM NaCl, 10 mM KCl, 5 mM MgCl$_2$, and 10% BSA).

### Subcellular fractionation

Subcellular fractionation was adapted from Gagnon et al (2014). Briefly, cells were lysed in ice-cold hypotonic lysis buffer (HLB; 10 mM Tris–HCl, pH 7.5, 10 mM NaCl, 3 mM MgCl$_2$, 0.3% NP-40, and 10% glycerol) for 10 min on ice. The cytoplasm was harvested by centrifugation (1,000$g$, 5 min) and the nuclear pellet was washed thrice in HLB. Nuclei were incubated in ice-cold modified Wuarin–Schibler buffer (MWS; 10 mM Tris–HCl, pH 7.5, 4 mM EDTA, 0.3 M NaCl, 1 M urea, and 1% NP-40) for 15 min on ice. The nucleoplasm was separated from the chromatin by centrifugation (1,000$g$, 5 min). The RNA in the cytoplasmic and nucleoplasmic fractions was ethanol-precipitated and subjected along with the chromatin pellet for RNA purification.

### RNA-seq analysis

Total RNA was isolated and polyA-enriched. After reverse transcription, the cDNA was fragmented, end-repaired, and polyA-tailed. Solexa sequencing adaptors were ligated, and adaptor-modified fragments were enriched by 10–18 cycles of PCR amplification. Quantity and the size of the sequencing library were accessed on a Bioanalyzer before sequencing on an Illumina NextSeq 500 platform. Sequencing reads from FASTAQ files were aligned the STAR Aligner version (Dobin et al, 2013) and quantified using rsem (Li & Dewey, 2011). The reference genome used for alignment was constructed using the mm10 fasta file and GRCm38.99 transcript table. Quantified values were further statistically evaluated using Bioconductor's DeSeq2 package (Love et al, 2014). Expression changes with an FDR < 0.05 were considered significant. Among them, genes with a stat < −2 or >2 were extracted as down- or up-regulated genes, respectively (Table S3).

### GO term enrichment analysis

GO enrichment used the Web-based PANTHER software (Mi et al, 2013). The deregulated genes enriching for GO terms of interest were extracted from the provided xml file and matched to their expression values using R.

### ChIRP-seq analysis

NSCs from 8 × 15-cm dishes were harvested 7 d after seeding and washed twice with PBS. Cells were cross-linked in 100 ml 1% glutaraldehyde for 10 min at RT. Cross-linking was quenched 125 mM

glycine for 5 min. Cells were pelleted at 1,000*g* for 5 min. ChIRP was performed according to Chu et al (2011). Cross-linked cells were washed twice in PBS and lysed in 2 ml ChIRP-lysis buffer (50 mM Tris–HCl pH 7.0, 10 mM EDTA, 1% SDS, 1 mM PMSF, 1× protease inhibitor, SuperaseIn 100 U/ml). Chromatin shearing by Bioruptor typically yielded fragments of 150–600 bp. Sheared chromatin was diluted with 4 ml ChIRP-hybridization buffer (50 mM Tris–HCl pH 7.0, 750 mM NaCl, 15% [m/v] formamide, 1 mM EDTA, 1% SDS, with protease and RNase inhibitors) and divided into two aliquots, which were hybridized with 100 pmol biotinylated "odd" and "even" probe sets, respectively, at 37°C for 4 h with continuous rotation. Then 1 mg of magnetic streptavidin bead suspension (Thermo Fisher Scientific) in ChIRP-Lysis buffer were added and incubated for 30 min at 37°C with continuous rotation. Beads were washed five times with 1 ml ChIRP Wash buffer (300 mM NaCl, 30 mM $Na_3$-citrate, 0.1% SDS, and 1 mM PMSF) for 5 min at 37°C. 90% of bead material was used for DNA isolation and 10% for RNA isolation. The enrichment of *RUS*, *TBP* mRNA, *MALAT*, and *XIST* was analyzed by RT-qPCR.

Isolated DNA was processed alongside an input chromatin sample. Ends were blunted with T4 DNA polymerase and polynucleotide kinase and an AMP was added. Solexa sequencing adaptors were ligated and adaptor-modified fragments were enriched by 10–18 cycles of PCR amplification. Sequencing libraries were size-selected on AMPure Beads (Beckman Coulter), quality-controlled on a Bioanalyzer (Agilent) and sequenced on an Illumina NextSeq-500 platform.

Sequencing reads from FASTQ files were aligned with bowtie2 (Langmead & Salzberg, 2012) to mm10. Multimapping reads were removed using samtools (Li et al, 2009). ChIRP peaks were called with MACS1.4 for both probe sets independently (Feng et al, 2012). The deeptools package was used to generate the bedgaph files (Ramírez et al, 2016). Bedtools (Quinlan & Hall, 2010) and python 2.7 matched even and odd bedgraph files into a single bedgraph file via the "take-lower" method. The experiment was performed in triplicates. Only peaks occurring in each even and odd sample and in all three data sets called with Bioconductor's GenomicRanges package (Lawrence et al, 2013) were considered valid *RUS* binding sites. The overlap demanded a minimal distance of 200 bp between the "even" and "odd" summit. Probe sequences within overlapping peaks were detected using Fimo (Grant et al, 2011) of the MEME software (Bailey et al, 2015) and removed using a cutoff of $p < 1 \times 10^{-8}$ before further analysis using GenomicRanges (Lawrence et al, 2013).

Filtered peaks were annotated with Homer (Heinz et al, 2010) using mm10 as reference genome (Table S4). The obtained annotation statistic was used to calculate the distribution of *RUS* peaks within promoter, intergenic, intron, and close to repetitive sites. The annotated neighboring genes of *RUS* peaks were considered putative *RUS* target genes. GO term enrichment of putative target genes used the Web-based PANTHER software (Mi et al, 2013). Next, putative target gene expression and changes upon in shRNA$^{CON}$ and shRNA$^{RUS}$ treatment on day 5 and 7 were extracted from the RNA-seq data using the R-package SummarizedExperiments and DeSeq2 (Table S4). Expression changes with an FDR < 0.05 were considered significant. Among them, genes with a stat < −2 or >2 were considered as down- or up-regulated genes. Expression values of both time points were merged, $log_2$-transformed, and ranked by

hierarchically clustering using the Euclidean distance method in R. Furthermore, the correlation between *RUS* and putative target gene expression was calculated using the Pearson correlation coefficient on both time points separately (Table S4).

## MS2 affinity purification of RUS interactors

Stable pools of Neuro2A cells expressing 5xMS2-tagged *RUS* were generated as follows. $5 \times 10^4$ Neuro2A cells were transfected with 5 µl pLenti-FRT virus and 2 d later selected in GlutMax, 8% FCS supplemented with 2 µg/ml puromycin and expanded. $10^6$ Neuro2A-FRT cells were seeded on a 10-cm culture dish. On the next day, cells were transfected with 15 µg plasmid DNA, consisting of a molar ratio of 1:6 (up to 1:9) of pcDNA5-lncRNA-5xMS2: pCSFLPe (encoding the flipase). Plasmids were diluted appropriately in 300 µl 150 mM NaCl and 15 µl JetPEI (2.6 µg/µl) and mixed. After 30 min equilibration at RT, the solution was added dropwise to Neuro2-FRT cells. 2 d later, cells were transferred to a new 10-cm dish and selected in GlutaMax 8% FCS, 2 µg/ml puromycin, and 600 µg/ml hygromycin. The medium was replaced every second day to remove cell debris. Colonies formed 7–10 d after transfection. They were harvested and further cultivated.

Nuclear extract from MS2-tagged *RUS*-expressing Neuro2A cells was prepared typically from $8 \times 10^7$ cells without dialysis, according to Dignam et al (1983). Extract preparation and MS2-affinity purification were carried out at 4°C. Cell pellets were suspended in 5 vol buffer A (10 mM Hepes, pH 7.9 at 4°C, 1.5 mM $MgCl_2$, 10 mM KCl, 0.5 mM DTT, and 200 U/ml RNAsin) and incubated for 10 min. Cells were homogenized with a Dounce tissue grinder. Nuclei were pelleted at 500*g* for 10 min, washed with five nuclear volumes (vol) buffer A, dissolved in one vol buffer C (20 mM Hepes, pH 7.9, 25% [vol/vol] glycerol, 0.42 M KCl, 1.5 mM $MgCl_2$, 0.2 mM EDTA, 0.5 mM PMSF, 0.5 mM DTT, and 200 U/ml RNAsin) and homogenized again with a Dounce tissue grinder. After gentle rotation for 30 min, chromatin was pelleted at 17,000*g* for 30 min. The supernatant was diluted with 1 vol buffer G (20 mM Hepes, pH 7.9, 20% [vol/vol] glycerol, 0.2 mM EDTA, 0.5 mM PMSF, 0.5 mM DTT, and 200 U/ml RNAsin) and used for affinity purification.

Standard MS2-affinity purification was carried out on supernatant containing 1 mg protein. To this, 760 pmol yeast t-RNA competitor and 120 pmol recombinant MS2BP-MBP (Jurica et al, 2002; Zhou & Reed, 2003) was added. After 2 h of gentle rotation, 50 µl equilibrated amylose resin (New England Biolabs) was added and incubation continued for 2 h. The resin was pelleted at 1,900*g* for 1 min and washed thrice with 900 µl buffer D (buffer G containing 0.1 M KCl and lacking RNasin) and thrice 900 µl buffer F (buffer D containing 1.5 mM $MgCl_2$).

RNA-interacting proteins were identified by mass spectrometry. Interacting proteins were eluted with 50 µg RNAse A in 80 µl buffer D at 37°C for 10 min. The resin was pelleted at 1,900*g* for 1 min at 4°C and the supernatant subjected to filter-aided sample preparation (Wiśniewski et al, 2009), and peptides were desalted using C18 StageTips, dried by vacuum centrifugation, and dissolved in 20 µl 0.1% formic acid. Samples were analyzed on a Easy nLC 1,000 coupled online to a Q-Exactive mass spectrometer (Thermo Fisher Scientific). 8 µl peptide solution per sample were separated on a self-packed C18 column (30 cm × 75 µm; ReproSil-Pur 120 C18-AQ,

1.9 µm, Dr. Maisch GmbH) using a 180-min binary gradient of water and acetonitrile supplemented with 0.1% formic acid (0 min, 2% B; 3: 30 min, 5% B; 137:30 min, 25% B; 168:30 min, 35% B; 182:30 min, 60% B) at 50°C column temperature. A top 10 DDA method was used. Full scan MS spectra were acquired with a resolution of 70,000. Fragment ion spectra were recorded using a 2 m/z isolation window, 75 ms maximum trapping time with an AGC target of $10^5$ ions.

The raw data were analyzed with the MaxQuant (version 2.0.1.0) software (Cox & Mann, 2008) using a one protein per gene canonical database of *Mus musculus* from UniProt (download : 2021-04-09; 21,998 entries). Trypsin was defined as protease. Two missed cleavages were allowed for the database search. The option first search was used to recalibrate the peptide masses within a window of 20 ppm. For the main search, peptide and peptide fragment mass tolerances were set to 4.5 and 20 ppm, respectively. Carbamido-methylation of cysteine was defined as a static modification. Acetylation of the protein N terminus as well as oxidation of methionine set as variable modifications. Match between runs was enabled with a retention time window of 1 min. Two ratio counts of unique peptides were required for LFQ.

Output files were further analyzed using the software Perseus (Tyanova et al, 2016). Proteins identified by site, reverse matching peptides and contaminants were removed and LFQ intensities were $\log_2$-transformed. Next, only protein groups with five out of five quantifications in one condition were considered for relative protein quantification. To account for proteins that were only consistently quantified in one condition, data imputation was used with a down-shift of 2 and a width of 0.2. A permutation-based FDR correction (Tusher et al, 2001) for multiple hypotheses was applied ($P$ = 0.05; s0 = 0.1). Proteins were considered enriched if the fold change was greater than two and the $P$-value less than 0.002 (Table S5).

### RNA immunoprecipitation

Protein A/G-Agarose beads (35 µl; Thermo Fisher Scientific) were blocked overnight with 1% BSA in buffer D. To nuclear extract from 5 × $10^6$ NSC 760 pmol yeast tRNA, 300 µg salmon sperm DNA and 4 µg Lbr antibody were added and incubated for 2 h at 4°C under gentle rotation. Anti-rabbit IgG was used as a negative control. The binding reaction was added to blocked Protein A/G–Agarose and incubated for 2 h at 4°C with gentle rotation. Protein A/G beads were sedimented, washed 5× with 900 µl buffer D, suspended in 800 µl Trizol, and subject to RNA extraction. RUS levels were analyzed by RT-qPCR analysis and compared against the IgG purification. The experiment was performed in triplicates and statistically evaluated by a one-tailed *t* test using Bonferroni *P*-value adjustment.

## Data Availability

The RNA-Seq and ChIRP Seq data from this publication were deposited to the Gene Expression Omnibus repository (https://www.ncbi.nlm.nih.gov/geo) with accessions GSE196487, GSE196527, respectively. The protein interaction AP-MS data can be found at the PRIDE repository (Perez-Riverol et al, 2022) (http://www.ebi.ac.

uk/pride/archive) with the accession PXD031664. Computer scripts are deposited on GitHub (https://github.com/MariusFSchneider/Schneider22).

## Supplementary Information

## Acknowledgements

We thank Aline Campos, Silke Krause, and Anna Berghofer for technical assistance, Tobias Straub for advice on bioinformatic analysis, Bianka Baying, Vladimir Benes (EMBL GeneCore), and Stefan Krebs (LAFUGA) for library preparation and sequencing, Magdalena Götz, Daniela Cimino, and Maroussia Hennes for providing mouse brain tissues and Christian Haass for his continued support. We are grateful to Sandra Schick, Marie Kube, and Rodrigo Villaseñor for critical reading of the manuscript. This work was funded by the Deutsche Forschungsgemeinschaft (DFG) within the framework of the Munich Cluster for Systems Neurology (EXC 2145 SyNergy– ID 390857198), grant BE1140/8-1 (to PB Becker), and the Adele Hartmann Programm of the LMU (to JC Scheuermann).

### Author Contributions

MF Schneider: conceptualization, data curation, visualization, methodology, and writing—original draft, review, and editing.
V Müller: investigation and methodology.
SA Müller: data curation, formal analysis, investigation, methodology, and writing—review and editing.
SF Lichtenthaler: funding acqisition, validation and writing—review and editing.
PB Becker: conceptualization, supervision, funding acquisition, writing—original draft, and project administration.
JC Scheuermann: funding acqisition, conceptualization, and project administration.

### Conflict of Interest Statement

The authors declare that they have no conflict of interest.

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
