## [Reviewer comments · Life Science Alliance]

LncRNA RUS shapes the gene expression program towards neurogenesis

Marius Schneider, Veronika Müller, Stephan Müller, Stefan Lichtenthaler, Peter B. Becker, and Johanna Scheuermann
DOI: 10.26508/lsa.202201504

Corresponding author(s): Peter B. Becker, Ludwig-Maximilians-University

Review Timeline:

Submission Date:	2022-04-25
Editorial Decision:	2022-04-26
Revision Received:	2022-05-09
Editorial Decision:	2022-05-10
Revision Received:	2022-05-13
Accepted:	2022-05-13

Transaction Report:

Please note that the manuscript was reviewed at Review Commons and these reports were taken into account in the decision-making process at Life Science Alliance.

April 26, 2022

Re: Life Science Alliance manuscript #LSA-2022-01504

Prof. Peter B. B. Becker
Ludwig-Maximilians-University
Biomedical Center Munich
Molecular Biology
Großhaderner Strasse 9
Planegg-Martinsried 82152
Germany

Dear Dr. Becker,

Thank you for submitting your manuscript entitled "LncRNA RUS shapes the gene expression program towards neurogenesis" to Life Science Alliance. We invite you to re-submit the manuscript, revised according to your Revision Plan.

Thank you for this interesting contribution to Life Science Alliance. We are looking forward to receiving your revised manuscript.

Sincerely,

Eric Sawey, PhD
Executive Editor
Life Science Alliance
<http://www.lsa-journal.org>

B. MANUSCRIPT ORGANIZATION AND FORMATTING:

Dear Dr. Sawey,

we thank you for the invitation to submit a revised version of our manuscript to be published in LSA. We revised our manuscript as proposed in the revision plan that you already approved. We very much appreciate the constructive comments and critique of the three reviewers that help us to mature our manuscript. The revisions led to multiple amendments to the text and two new sets of data, now presented in Figure 1E and 4C).

In the following we address all reviewer comments point by point. In the revised manuscript all textual changes are marked in red (quotations underlined below).

Reviewer #1

(summary). The manuscript gives an initial insight into the role of the lncRNA RUS in neural stem cell differentiation. Overall, this is a potentially interesting study that can be strengthened by a set of experiments listed below.

Response: We are grateful for the constructive feedback.

1. Sub-cellular localization and abundance of RUS: Sub-cellular localization of RUS was shown by fractionation experiment and/or single cell in situ hybridization. The authors should show their single cell in situ hybridization data (the current status in the manuscript is "data not shown") as it can provide additional information about the abundance of RUS in the cell (i.e. approximate number of RUS molecules per cell). This point is particularly important as the authors find >120 binding sites of RUS to chromosomal regions.

Response: We infer the subcellular localisation of RUS by subcellular fractionation, Chromatin isolation by RNA purification (ChIRP) and by affinity purification of numerous chromatin proteins. We also mentioned supporting fluorescence in situ hybridization (FISH) data as 'not shown'. Reviewing these data once more, we conclude that they are, while supportive, of qualitative rather than quantitative nature and cannot stand on their own as a figure. Because we are currently unable to engage in a larger FISH study, we decided to remove the sentence 'An RNA-FISH (fluorescence-in-situ-hybridization) experiment confirmed the nuclear localization (data not shown)'. We think that the nuclear localisation is well documented by the experiments listed above. Furthermore, we now added quantitative RT-PCR data monitoring expression of *RUS* and *nestin* in mouse cortical and hippocampal tissues and different development stages (E14, E18, P3, P8 and adult; new Fig 1E). We added the following description of the experiment on page 5 in the result section: "We further explored the spatio-temporal expression of *RUS* isoform 1 in the developing mouse brain. Quantitative RT-PCR analyses of *RUS-1* transcripts in cortex and hippocampus of different developmental stages (embryonic days E14 and E18, post-natal days P3 and P8 as well as adult animals) showed that *RUS-1* expression increased during cortical development and peaked on P3 when Nestin, a marker for neural precursor cells dropped. A reciprocal expression pattern was observed in the hippocampus."

2. Conservation and translation of putative ORF "Since the corresponding peptides are not listed in published mass spec data (PeptideAtlas), we assume that RUS functions as a lncRNA". No evidence has been provided for the noncoding function of RUS. This point can be addressed by using minimally invasive CRISPR-Cas9 strategies for inactivation of potential ORF and testing if the cells proliferate normally or show cellular phenotype resembling RUS depletion.

Response: This nice experiment suggested by the referee is currently beyond our technical capabilities, at least with the time and resources at our disposition. We think that our study provides strong support that RUS functions as an RNA. However, at this point we cannot formally exclude that some peptides may be translated under special conditions. Likewise, we cannot exclude that the long RNA is processed under some circumstances

and arising miRNA may contribute to the functional effects. We have been unable to find such peptides or miRNAs in public repositories (<http://www.peptideatlas.org/>, <https://www.mirbase.org/>).

To proactively alert the reader to the fact that we cannot exclude roles of encoded peptides and miRNAs, we modified the sentence in question on page 3 as follows: 'Although the corresponding peptides are not listed in the comprehensive peptide repository (<http://www.peptideatlas.org/>), we cannot exclude a function for a hypothetical polypeptide encoded by this small ORF. Likewise, we cannot exclude that RUS is processed to miRNAs (<https://www.mirbase.org/>) contributing to its functionality'. In addition, we now mention these formal possibilities more explicitly in the discussion on page 9.

3a. RUS inhibition and rescue experiments: It appears that only one shRNA was tested for RUS inhibition. The use of a single lncRNA perturbation method and the usage of only one shRNA, which is also prone to off-target effects, limits the robustness of the conclusions. At least, an additional shRNA should be included which is standard in the field.

Response: We share the reviewer's concern about potential off-target effects of shRNAs. Unfortunately, most of the shRNAs we tested did not lead to efficient depletion of RUS. The second-best shRNA we found targets exon 4 and hence an isoform of lower abundance. The overall reduction of RUS was lower and consequently, the effects milder. Therefore, we paid particular attention to designing a meaningful rescue experiment. This controlled rescue experiment reverses important aspects of the phenotype lending credibility to the notion of a causal role of RUS in eliciting the shRNA phenotype.

3b. The authors should clarify how the rescue experiment was carried out i.e. expression of the mature RUS transcript, which of the RUS isoforms etc.

Response: We now explain more explicitly how the rescue experiment was done (Materials and Methods). Specifically, we added the important information that the rescue construct expresses RUS isoform 1, which is targeted by shRNA^{RUS}.

4. RUS interactors and their function in cell proliferation. In an elegant pull-down experiment, the authors have defined the protein interactome of RUS. It is a well-designed and executed experiment. This is one of the strongest parts of the study. However, the authors did not go further in their functional analyses of the protein interactors to determine which of them contribute to RUS functionality. At the current state, this part of the manuscript provides only an initial insight into the potential molecular mechanism of RUS action remaining speculative. Having a robust cellular read-out such as decreased cell proliferation and increased apoptosis, a depletion of individual RUS interaction partners may clarify their contribution to the cellular phenotype.

Response: We thank the reviewer for the appreciation. We agree that our findings now allow multiple follow-up experiments to establish causalities and mechanism. Unfortunately, we are unable to follow up these leads and must leave these opportunities to others in the research community.

5. Slitrk3 expression upon RUS inhibition. Have the authors checked expression levels of Slitrk3 upon RUS inhibition? Although rescue experiments rather an in-trans action of RUS, it will be still important to test Slitrk3 expression levels upon RUS depletion.

Response: We indeed checked and added a sentence on page 6, indicating this fact: "Consistent with findings that many lncRNAs regulate the expression of genes in the vicinity of their sites of transcription, the expression of the Slitrk3 was significantly reduced after RUS depletion (Table S3). In addition, the depletion of RUS massively affected the transcriptome, arguing that RUS also acts in trans."

Significance: The study is potentially interesting for the noncoding RNA field. Technically, several parts of the study are very well-done (ie. rescue experiments, ChIRP, pull-down experiments). The significance of the study for the field will increase if the authors were able to provide functional data for the RUS interactors (see my major point #4). State what audience might be interested in and influenced by the reported findings. Given that I am an expert in ncRNA biology including their functions and mechanisms of action, I have sufficient expertise to evaluate the study.

Response: We agree with this assessment.

Reviewer #2

(summary). In general, this study is quite novel, carefully designed and organized. Similarly, the manuscript is very well written, concise and clear. Most of the presented data are convincing and well presented.

My main issue (major concern) is that, although the authors manage to reveal many molecular, cellular and mechanistic aspects of RUS function in NSCs, they are not investing some extra time/effort to bring all this knowledge together. I understand that, at the end, it turned out that RUS has an intricate and complex network of gene and protein interactions that make such synthesis extremely difficult and maybe impossible at this stage. However, I would like to suggest to continue a little further, by investigating the ability of couple of interacting proteins/complexes (identified by the authors) to affect the RUS function. For example, is brd2 or Lbr knockdown able to abolish/impair RUS function?

Response: We thank the reviewer for the appreciation. We agree that our findings now allow multiple follow-up experiments to establish causalities and mechanism. Unfortunately, we are unable to follow up these leads and must leave these opportunities to others in the research community.

Moreover, some extra comments listed below should be addressed:

1. It would be very supportive for the expression analyses of this study, if in situ detection of RUS is performed in mouse or human neural cells and/or tissues. Such analysis is completely missing from this nice study.

Response: Please see our response to point 1 of reviewer 1.

2. In Fig. 2, it would be very helpful for the readers if the authors indicate on the Figure that this analysis has been performed in mouse cortical NSCs.

Response: We thank the reviewer for alerting us to this omission. We now added this important information.

3. At the end of Fig. S2, the authors conclude that RUS affects both proliferation and apoptosis. By this analysis is not clear whether the effect of RUS is on both cellular events, or whether by affecting apoptosis the proliferating cells (or a proliferating sub pool) is reduced. I understand that latter on the study the authors identify cell cycle genes such as ccnd1, ccna2, cdk1 etc, as RUS-regulated genes, suggesting that this conclusion is probably correct. But at this early point is a bit confusing.

Response: We think that RUS depletion primarily affects the differentiation program, which involves amplification of intermediate precursor cells (and thus transient proliferation cues) followed by NSC differentiation. Perturbation of this program may trigger an apoptotic signature. This is spelled out in the discussion. We added a summary statement on page 5: "Our subsequent analysis suggested that both processes contribute to cell loss. To explore proliferation effects ..."

4. Similarly, it is not absolutely clear that the effect of RUS on neuronal differentiation is due to an impairment of differentiation program or an indirect effect due to increased apoptosis. I suggest to add a short discussion of this issue in the Discussion section.

Response: We amended the discussion on page 10 as follows: "However, it is also possible that increased apoptosis after RUS depletion impaired neuron formation."

5. In Fig 3 and Fig 4, I feel that it would be very helpful if the authors invest some time to confirm the RNA-seq data. This is particularly valid for the direct gene targets of RUS, i.e. *Kcna1*, *Dclk2*, etc. Thus, I would like to perform real time RT-qPCR assays for these targets, but most importantly protein analyses (e.g. Western blots and/or IF analyses). It is quite critical to examine whether these gene expression changes identified with a genome wide method correspond to changes at the protein levels.

Response: We are grateful for the suggested validation of the RNA-seq data. We now provide new RT-qPCR quantification for 6 target genes and present the results on page 8 (new Fig 4C) as follows: "Quantification of the mRNA levels of *Bin1*, *Kcna5*, *Arid1b*, *Dclk2*, *Dpp9*, and *App* by RT-qPCR confirmed the increase in these target genes after RUS depletion on day 7 (Fig 4C)." To claim the space we moved previous Figs 4 A, B to S4.

6. Finally, it would be extremely supportive of the main conclusions of this study, if authors manage to provide a short of *in vivo* confirmation of RUS action in NSCs (e.g. by *in utero* electroporation in the mouse cortex).

Response: This nice experiment suggested by the referee is currently beyond technical capabilities, at least with the time and resources at our disposition.

Significance: In general, this study is quite novel, carefully designed and organized. Similarly, the manuscript is very well written, concise and clear. Most of the presented data are convincing and well presented.

Response: We are grateful for this very positive evaluation.

Reviewer #3

(summary). Major comments:

1. The authors set out by stating in the abstract and introduction that the evolution of brain complexity correlates with an increased expression of long non-coding (lnc) RNAs in neural tissues. However, their study has nothing to do with brain evolution as they specifically looked for lncRNAs conserved between mouse and humans (criterion 3).

Response: The reviewer is, of course, correct. Nevertheless, we kept the statement about brain evolution, because this is a fascinating correlation that illustrates lncRNAs as important regulatory principle for fast evolution of complexity. However, the next sentence in the introduction states: "Brain-specific lncRNAs tend to be more evolutionary conserved between orthologues than lncRNAs expressed in other tissues and their genes often reside next to protein-coding genes involved in neuronal development or brain function processes (Ponjavic et al, 2009)". This sentence then introduces two aspects that are relevant for our work: evolutionary conservation and synteny.

2. It is not clear - and this part needs re-writing - that the authors first meta-analyzed *in silico* the human transcriptomics data of Ziller et al (2015). This should be explicitly stated at the beginning of the Results section. It is also not clear against which mouse database they compared the human data to come across RUS.

Response: We are sorry if the compact text led to confusion. We now rewrote the paragraph in question as follows: 'To identify novel, functionally relevant lncRNAs in the context of neurogenesis, we took advantage of prior work of Ziller et al., who profiled transcription during differentiation of human embryonic stem cells along the neural lineage (Ziller et al., 2015). Their data include transcriptome profiles of hESC-derived neural progenitors: neuroepithelial cells (NE), early, mid and late radial glia cells (ERG, MRG, LRG, respectively) and their in vitro differentiated counterparts (Ziller et al, 2015). We evaluated 553 candidate lncRNA transcripts according to the following criteria. They should 1) only be expressed in neural tissues, 2) be dynamically regulated during the differentiation of neural precursor cells and 3) be conserved between mouse and humans (Fig 1A).'

We also amended the legend to figure 1A to indicate which databases were used for initial data mining.

3. It is not clear if the authors used the neurosphere assay in their study. The papers referenced have correctly used the neurosphere assay at first, to make sure that proliferating self-renewing cells are isolated from the brain and have then proceeded to dissociating and seeding these precursor cells as adherent cultures. Have the authors used the same procedure? As far as I can see in the Methods section, this is not the case and compromises their data.

Response: We thank the reviewer for the opportunity to clarify this important experimental aspect. We used similar isolation procedure but kept the cells as adherent monolayer cultures. Isolated cells were passaged several times prior experiments to ensure the isolation of proliferating self-renewing cells. This is now explicitly stated in the Method section (Cultivation and differentiation of primary neural stem cells).

4. There is no evidence in which type of cells endogenous RUS is expressed during mouse brain development and in the adult. In situ hybridization data should be provided on tissue sections from different ages to support the in vitro data and conclusions.

Response: Please see our response to point 1 of reviewer 1.

5. It is very surprising that upon RUS knockdown there is such a vast effect on the transcriptome. According to the data presented, it seems that 46% of all transcribed genes are differentially expressed at day 5 and 60% at day 7. How is it possible that perturbation of a single lncRNA (to only 50% of its expression - incomplete depletion) has such a profound effect? Have the authors considered the possibility that RUS might act through miRNA regulation?

Response: We agree with the reviewer that the effect of RUS depletion is remarkable. At this point we cannot formally exclude that RUS is processed under some circumstances and arising miRNA may contribute to the functional read-out, and it is beyond our capabilities to experimentally address the issue. Likewise, we cannot exclude that some peptides may be translated under particular conditions. We have been unable to find such peptides or miRNAs in public repositories.

To proactively alert the reader to the fact that we cannot exclude roles of encoded peptides and miRNAs, we modified the sentence 'Since the corresponding peptides are not listed in published mass spec data (PeptideAtlas), we assume that RUS functions as a lncRNA' on page 3 as follows: 'Although the corresponding peptides are not listed in the comprehensive peptide repository (<http://www.peptideatlas.org>), we cannot exclude a functional for a hypothetical polypeptide encoded by this small ORF. Moreover, we cannot exclude that RUS is processed to miRNAs (<https://www.mirbase.org/>) contributing to its functionality'.

We also added a sentence to the discussion on page 9 to alert the reader to this formal possibility: "However, at this point we cannot exclude the formal possibility that a fraction of RUS is processed to functionally relevant miRNAs."

At this point, we suggest that RUS may affect multiple pathways governing primarily the amplification of intermediate precursor cells (proliferation control), neurogenic differentiation (differentiation program) and, as a secondary effect of perturbation, the apoptotic program. Since the primary cells in our cultures are not synchronous, these distinct regulatory programmes and their modulation are monitored in parallel. Of course, secondary effects upon deregulating master regulators (of the kind shown in Fig 3C, and 4B) are scored as well.

6. At day 5 upon RUS knock-down, the transcriptomic analysis implies that apoptosis genes are up-regulated while cell cycle and neurogenesis/neuronal differentiation genes are down-regulated. However, at 7 days neurogenesis/neuronal differentiation genes are up-regulated. This contradicts the loss-of-function data presented in Fig. 2 where neuronal differentiation is clearly compromised at day 5. How do the authors reconcile these conflicting results? Moreover, the pattern of endogenous RUS expression implies a role in promoting neuronal differentiation which is compatible with the data in Fig. 2. In the absence of evidence for direct effects, I think that the authors should refer to their claims regarding RUS function as speculative.

Response: As mentioned in the response to point 5, we think RUS affects several regulatory pathways in an asynchronous primary cell culture. Our current interpretation is that RUS-depleted cells are locked in their differentiation stage. We tried to highlight this by presenting the expression of proneuronal genes such as Rest and that were significantly upregulated after RUS depletion and contributed to the overrepresentation of GOs such as neuron differentiation for overexpressed genes. Although the expression pattern of RUS suggests for a function for neuron differentiation only, our data suggest that RUS already has an early function in the intermediate stage of neuron formation.

We now amended the discussion on pages 10 as follows: We removed: "We propose that RUS depletion locks neuronal precursors in an intermediate state towards neuronal differentiation, with arrested cell cycle." We also toned down our wording in the discussion to alert the reader about the speculative nature of our interpretation by stating: 'Although the pattern of endogenous RUS expression and the observation that neuron formation was impaired after RUS depletion suggest a role of the lncRNA in promoting neuronal differentiation, RNA-seq and GO analysis showed a significant upregulation of neuronal differentiation genes on day 7 after RUS depletion. Such conflicting results may be a consequence of upregulation of proneuronal genes such as Notch2 and Rest after RUS depletion. We speculate that RUS depletion locks neuronal precursors in an intermediate state towards neuronal differentiation, with arrested cell cycle. The activation of pro-apoptotic genes may result from perturbed cell identity. However, it is also possible that increased apoptosis after RUS depletion impaired neuron formation.'

7. Although RUS is localized next to the neurodevelopmental gene Slitrk3 that is critical for inhibitory GABAergic synapse development, there is no effort made to see if RUS affects SlitRK3 transcription in any way.

Response: We indeed have checked and now state the result on page 8: "Consistent with findings that many lncRNAs regulate the expression of genes in the vicinity of their sites of transcription, the expression of the Slitrk3 was significantly reduced after RUS depletion (Table S3)".

Significance: It is becoming increasingly evident that most of the transcribed genome corresponds to non-coding genes, including lncRNAs and miRNAs which are important in regulating gene expression in a co- and post-transcriptional manner. The brain in particular contains a large number of lncRNAs and miRNAs for only a few of which the function is known. Only a small fraction of lncRNAs involved in neurodevelopment and brain function has been studied in detail. Therefore the identification and functional characterization of lncRNAs involved in brain development and function is of great significance. The present manuscript comes to add to present literature the characterization of a novel lncRNA that may be

involved in neuronal differentiation, which should be of interest to neuroscientists and developmental biologists. I have a long-standing interest in neurogenesis and neuronal differentiation using in vivo mouse models and in vitro systems, including HUES-derived NPCs and neurons as well as their transcriptomics analysis at different stages of differentiation.

Response: We are grateful for this expert assessment of our paper.

We hope that the manuscript as revised will be acceptable for publication in 'Life Science Alliance'.

May 10, 2022

RE: Life Science Alliance Manuscript #LSA-2022-01504R

Prof. Peter B. B. Becker
Ludwig-Maximilians-University
Biomedical Center Munich
Molecular Biology
Großhaderner Strasse 9
Planegg-Martinsried 82152
Germany

Dear Dr. Becker,

Thank you for submitting your revised manuscript entitled "LncRNA RUS shapes the gene expression program towards neurogenesis". We would be happy to publish your paper in Life Science Alliance pending final revisions necessary to meet our formatting guidelines.

- please add a running title to our manuscript system
- please add the Twitter handle of your host institute/organization as well as your own or/and one of the authors in our system
- please add a callout for Figure S4E and Table S1 to your main manuscript text; please double-check your figure callouts for Figure 4; you have a callout for Figure 4D, but this is not in the legend or the figure

A. FINAL FILES:

B. MANUSCRIPT ORGANIZATION AND FORMATTING:

**Submission of a paper that does not conform to Life Science Alliance guidelines will delay the acceptance of your

manuscript.**

The license to publish form must be signed before your manuscript can be sent to production. A link to the electronic license to publish form will be sent to the corresponding author only. Please take a moment to check your funder requirements.

Sincerely,

May 13, 2022

RE: Life Science Alliance Manuscript #LSA-2022-01504RR

Prof. Peter B. B. Becker
Ludwig-Maximilians-University
Biomedical Center Munich
Molecular Biology
Großhaderner Strasse 9
Planegg-Martinsried 82152
Germany

Dear Dr. Becker,

Thank you for submitting your Research Article entitled "LncRNA RUS shapes the gene expression program towards neurogenesis". It is a pleasure to let you know that your manuscript is now accepted for publication in Life Science Alliance. Congratulations on this interesting work.

DISTRIBUTION OF MATERIALS:

Again, congratulations on a very nice paper. I hope you found the review process to be constructive and are pleased with how the manuscript was handled editorially. We look forward to future exciting submissions from your lab.

Sincerely,
